# HDAC3 is a molecular brake of the metabolic switch supporting white adipose tissue browning

Alessandra Ferrari [1,5], Raffaella Longo[1], Erika Fiorino[1], Rui Silva[1], Nico Mitro [1], Gaia Cermenati[1], Federica Gilardi[2], Béatrice Desvergne[2], Annapaola Andolfo[3], Cinzia Magagnotti [3], Donatella Caruso[1], Emma De Fabiani[1], Scott W. Hiebert[4] & Maurizio Crestani [1]

White adipose tissue (WAT) can undergo a phenotypic switch, known as browning, in response to environmental stimuli such as cold. Post-translational modifications of histones have been shown to regulate cellular energy metabolism, but their role in white adipose tissue physiology remains incompletely understood. Here we show that histone deacetylase 3 (HDAC3) regulates WAT metabolism and function. Selective ablation of *Hdac3* in fat switches the metabolic signature of WAT by activating a futile cycle of de novo fatty acid synthesis and β-oxidation that potentiates WAT oxidative capacity and ultimately supports browning. Specific ablation of *Hdac3* in adipose tissue increases acetylation of enhancers in *Pparg* and *Ucp1* genes, and of putative regulatory regions of the *Ppara* gene. Our results unveil HDAC3 as a regulator of WAT physiology, which acts as a molecular brake that inhibits fatty acid metabolism and WAT browning.

[1] Dipartimento di Scienze Farmacologiche e Biomolecolari, Università degli Studi di Milano, Milano 20133, Italy. [2] Centre Intégratif de Génomique, Université de Lausanne, Lausanne 1015, Switzerland. [3] ProMiFa, Protein Microsequencing Facility, San Raffaele Scientific Institute, Milano 20132, Italy. [4] Department of Biochemistry, Vanderbilt University School of Medicine, Nashville, TN 37232, USA. [5] Present address: Department of Pathology and Laboratory Medicine, University of California Los Angeles, Los Angeles, CA 90095, USA. Correspondence and requests for materials should be addressed to M.C. (email: maurizio.crestani@unimi.it)

White (WAT) and brown (BAT) adipose tissues differ in their morphology and functionality and are marked by distinct metabolic signatures. However, upon specific environmental cues (for example, cold exposure), WAT switches to a brown like-phenotype in a phenomenon called browning[1–3]. The identification of factors that may contribute to control browning is still a matter of intense investigations. Several studies demonstrated that HDACs participate in metabolic regulation of different organs, especially highlighting a pivotal role played by histone deacetylase 3 (HDAC3) as modulator of metabolic reprogramming in liver[4, 5] and heart[6, 7]. Our previous studies demonstrated that chemical inhibition of class I HDACs with MS-275 ameliorates obesity and glucose tolerance in *db/db* and diet-induced obese mice, by reshaping adipose tissue metabolism and via induction of WAT browning[8, 9]. In this regard, it has been shown that HDAC1 and HDAC2 play a role in adipose tissue physiology[10]; furthermore, genetic ablation in adipose tissue of NCoR1[11], a co-repressor interacting with HDAC3, improved adipose tissue functionality, and insulin resistance[12].

Several groups reported evidences about a link between metabolism and post-translational modifications of chromatin[4, 5, 13–16], sparking new interest in the role of epigenome modifiers in determining the metabolic signature of specific tissues. However, the links between metabolic phenotype and chromatin modifications (for example, histone acetylation), and the underlying regulators of the epigenome dynamics in adipose tissue have started to emerge only recently.

By using a combination of transcriptomic, metabolomic, and proteomic approaches coupled with genetic manipulation of mouse and cellular models, we show that HDAC3 regulates the metabolic phenotype of WAT by acting as a molecular brake of the cellular metabolic rewiring that sustains WAT browning.

## Results

**Differential expression of metabolic genes in BAT vs. WAT.** To characterize the phenotype of fat specific *Hdac3* KO mice, we first verified that floxed mice, here used as controls for comparison vs. H3atKO mice, display the typical molecular signature underlying the distinct morphological and functional features of WAT and BAT. To this end, we conducted a focused gene expression profiling of inguinal WAT (IngWAT) and BAT in floxed C57Bl/6J mice. As expected, this analysis showed that BAT is

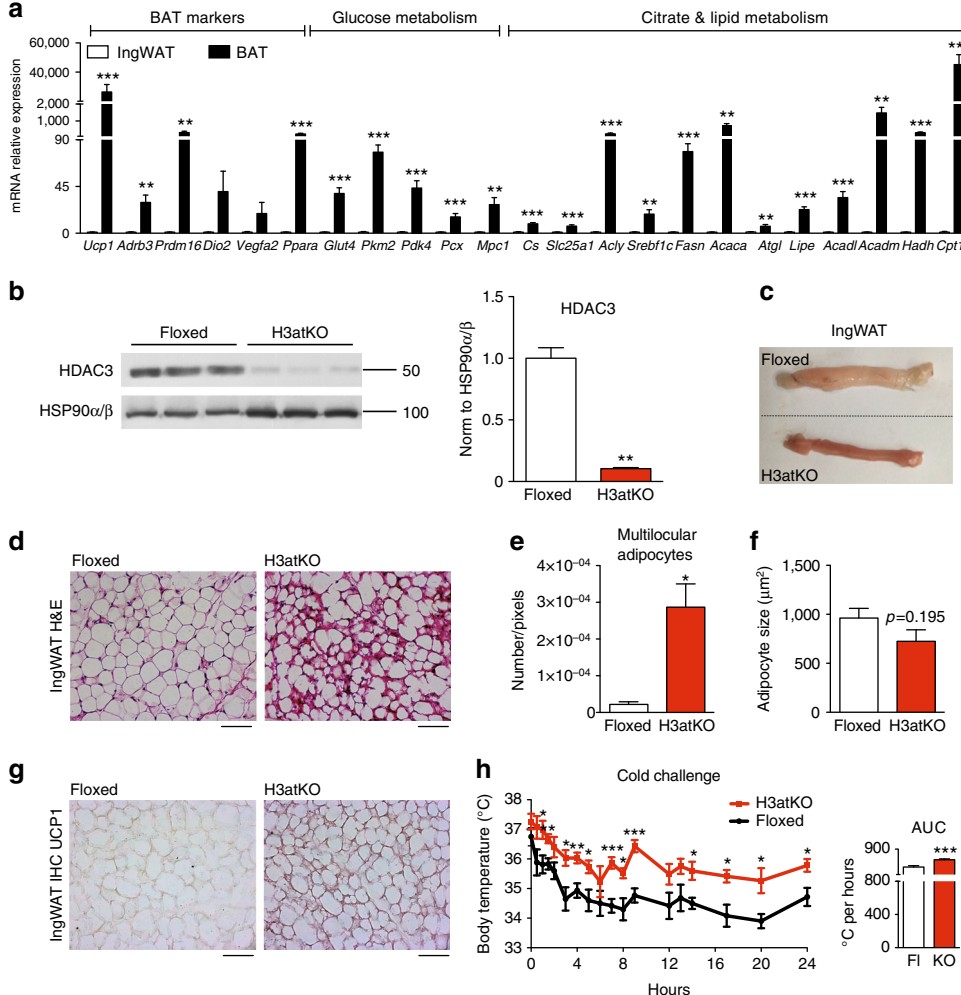

**Fig. 1** Adipose ablation of *Hdac3* results in browning of WAT in mice on LFD. **a** Gene expression analysis of BAT markers and of genes involved in glucose and lipid metabolism in brown and inguinal white adipose tissues (IngWAT) (*n* = 11); **b** Western blot analysis of HDAC3, showing the deletion of the protein in IngWAT of *Hdac3* knock out (H3atKO) mice vs. floxed mice (*n* = 3); **c** Representative image of IngWAT from floxed and H3atKO mice; **d** Hematoxylin–eosin staining of IngWAT from floxed and H3atKO mice, scale bar is 100 μm.; **e**, **f** Quantification of multilocular adipocytes and adipocyte size (*n* = 3 per group); **g** Immunohistochemical analysis of UCP1 IngWAT of floxed and H3atKO mice; **h** Body temperature of floxed and H3atKO mice exposed to 4 °C for 24 h (*n* = 6 per group). Data are presented as mean ± s.e.m. Statistical analysis: Student's *t*-test, *$p < 0.05$, **$p < 0.01$, and ***$p < 0.001$

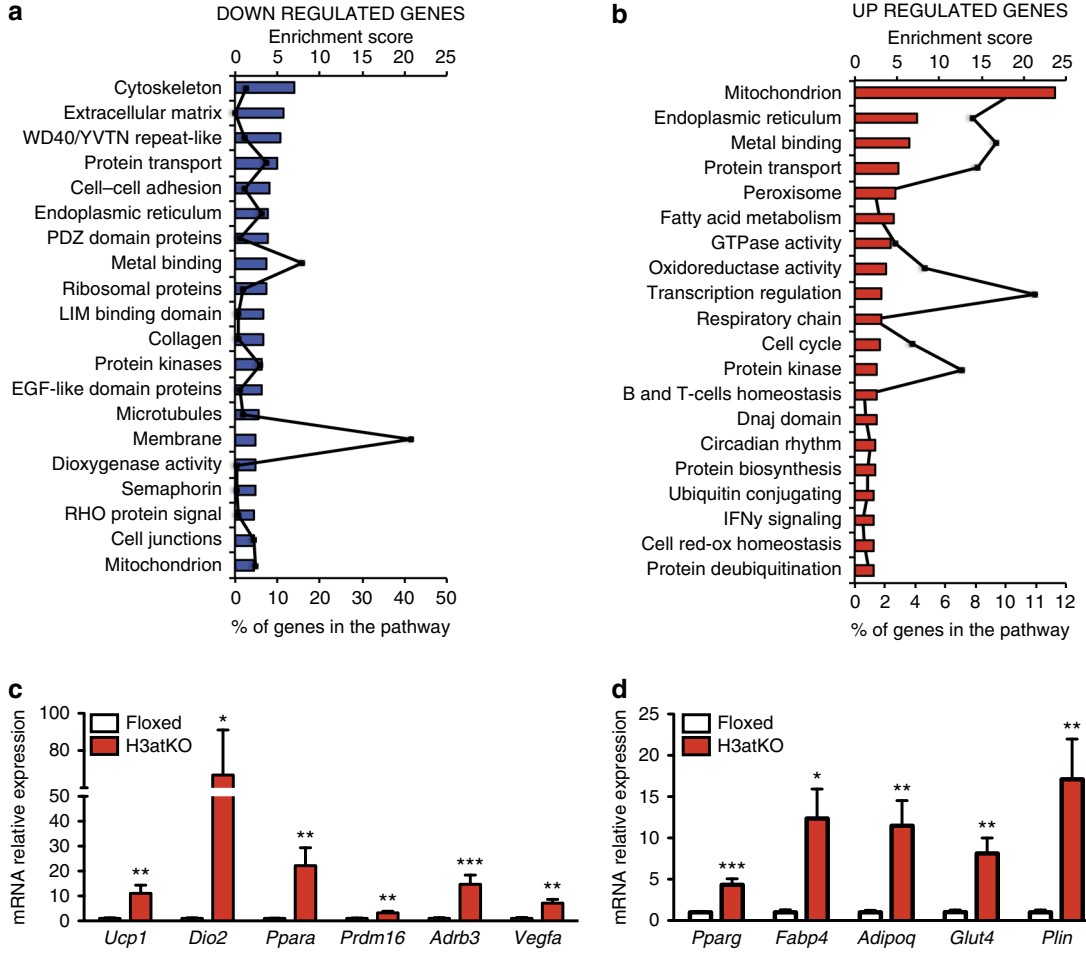

**Fig. 2** Differential transcriptome profile in IngWAT of floxed vs. H3atKO mice. **a** Down-regulated pathways in IngWAT of H3atKO mice; **b** Upregulated pathways in IngWAT of H3atKO mice (*n* = 3 per group, where each sample represents a pool from 2 individual mice); **c**, **d** qPCR gene expression analysis of browning genes and adipose markers in IngWAT of floxed and H3atKO mice (*n* = 9–11 per group). Data in **c** and **d** are presented as mean ± s.e.m. Statistical analysis: Student's *t*-test, *\*p* < 0.05, *\*\*p* < 0.01, and *\*\*\*p* < 0.001

characterized by remarkably high mRNA levels of brown fat specific genes such as those encoding for uncoupling protein 1 (*Ucp1*), β3-adrenergic receptor (*Adrb3*), deiodinase 2 (*Dio2*), vascular endothelial growth factor 2a (*Vegfa2*), PR domain containing 16 (*Prdm16*) and peroxisome proliferator-activated receptor α (*Ppara*) (Fig. 1a). However, this comparison revealed that BAT also showed exceptional upregulation of several genes involved in lipid, glucose, and citrate metabolism (Fig. 1a), highlighting brown fat as an organ with "high metabolic rate", with a concurrent activation of lipolytic/fatty acid (FA) β-oxidation and lipogenic gene programs in BAT, to meet the energy needs related to thermogenic function[17–19]. Overall, the comparative analysis of gene expression confirmed the differential metabolic signature of brown vs. WAT in this strain of mice. We thus used the fat specific *Hdac3* KO mouse model to investigate the role of HDAC3 in the differential metabolic phenotype of brown vs. WATs. The transcriptional analysis of *Hdac3* mRNA revealed a threefold higher expression in BAT as compared to IngWAT, in floxed control mice (Supplementary Fig. 1a).

**Adipose ablation of HDAC3 induces browning of white fat.** Several studies demonstrated that HDACs participate in metabolic regulation of different organs[4–7]. We previously showed that chemical class I HDAC inhibitor MS-275 improves

obesity and glucose tolerance in *db/db* and diet-induced obese mice, via induction of WAT browning[8, 9]. Since it has been demonstrated that genetic ablation of NCoR1 in adipose tissue[11], a co-repressor interacting with HDAC3, improved adipose tissue functionality and insulin resistance[12], we decided to investigate the role of HDAC3 in adipose tissue physiology and metabolism. To this end, we deleted *Hdac3* in adipose tissues by breeding *Hdac3* floxed mice with mice expressing the Cre recombinase under the control of the adiponectin gene promoter (*Adipoq*). Selective deletion of *Hdac3* in adipose tissues (H3atKO) was confirmed by PCR and western blot (Fig. 1b, Supplementary Fig. 1b, Supplementary Fig. 2a, b, Supplementary Fig. 3a, b). Interestingly, we observed that, despite no differences in total body weight gain (Supplementary Fig. 1c), H3atKO mice showed reduction of epididymal WAT (EpiWAT; Supplementary Fig. 2c), accompanied by a mild, though not statistically significant, reduction of IngWAT (Supplementary Fig. 1d). Surprisingly, IngWAT appeared reddish (Fig. 1c) and hematoxylin/eosin staining showed a clear difference of cellular morphology, characterized by smaller, though not statistically significant, cell size and by the presence of multilocular adipocytes (Fig. 1d–f). The immunohistochemical analysis of IngWAT revealed strong increase of UCP1 staining (Fig. 1g). Most importantly, when exposed to cold H3atKO mice were able to maintain body temperature to values higher than those of floxed mice (Fig. 1h), thus suggesting browning of WAT in H3atKO mice, which would

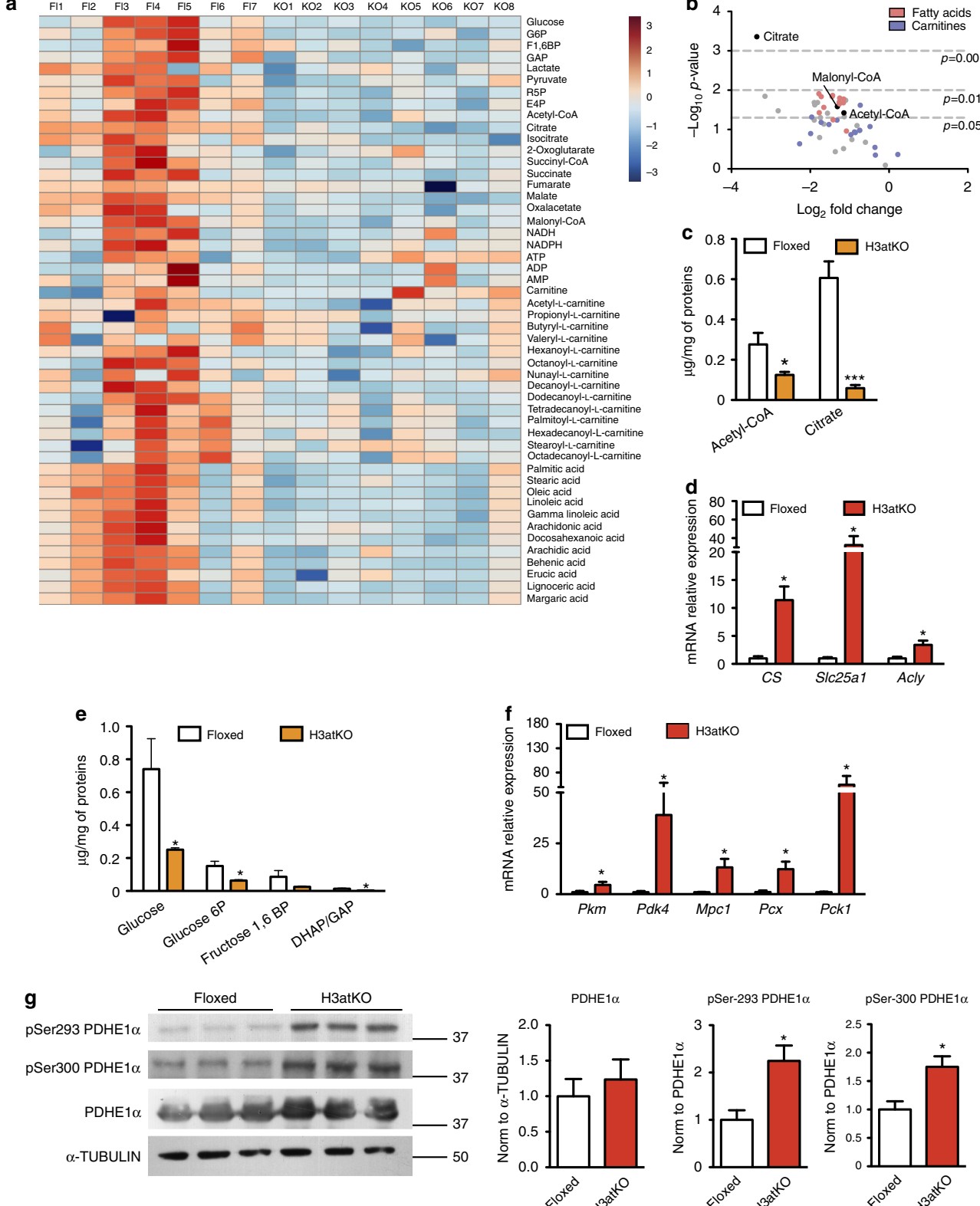

**Fig. 3** Adipose ablation of *Hdac3* remodels WAT metabolism. **a** Heatmap of IngWAT metabolome in floxed and H3atKO mice (*n* = 7–8 per group); **b** Volcano plot of IngWAT metabolome in floxed and H3atKO mice (*n* = 7–8 per group). **c** Levels of acetyl-CoA and citrate in IngWAT of floxed and H3atKO mice (*n* = 7–8 per group); **d** Expression analysis of genes involved in citrate mitochondrial export in IngWAT of *Hdac3* floxed and H3atKO mice (*n* = 9–11 per group); **e** Levels of glycolysis intermediates in IngWAT of floxed and H3atKO mice (*n* = 7–8 per group); **f** Expression analysis of genes involved in glucose metabolism in IngWAT of *Hdac3* floxed and H3atKO mice (*n* = 9–11 per group); **g** Western blot and quantification of pSer-293 PDHE1α, pSer-300 PDHE1α, and PDHE1α, α-Tubulin in IngWAT of *Hdac3* floxed and H3atKO mice (*n* = 3 per group). Data are presented as mean ± s.e.m. Statistical analysis: Student's *t*-test, *$p < 0.05$, and ***$p < 0.001$

contribute to improve the response to cold. RNA sequencing was performed in IngWAT from a subset of floxed and H3atKO animals fed standard diet for an initial characterization of transcriptome. Interestingly, we found that several genes were differentially regulated in IngWAT in response to adipose ablation

of HDAC3 (Fig. 2a, b). Among upregulated transcripts we found genes belonging to functional annotations related to mitochondria, FA metabolism and Krebs cycle, indicating activation of exceptional oxidative program in white adipocytes upon HDAC3 deletion. Of note, several other genes were down-regulated in

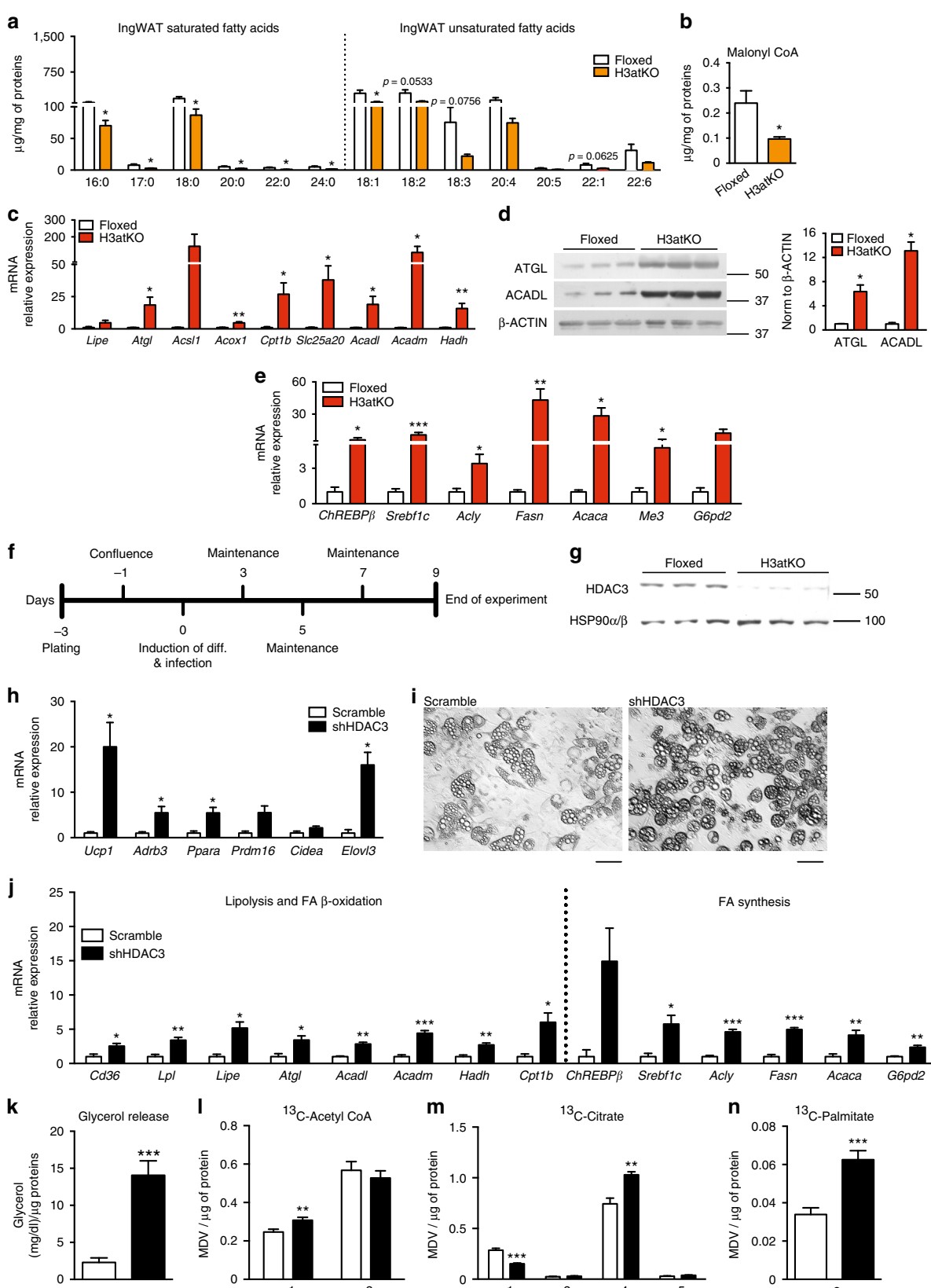

H3atKO mice, indicating that ablation of this transcriptional repressor did not de-repress globally the entire transcriptome (1429 upregulated transcripts and 2246 down-regulated transcripts). Based on the results of RNA sequencing we decided to conduct quantitative PCR (qPCR) analysis on specific panels of genes, playing a role in adipose physiology and in WAT browning. We confirmed increased expression of classical brown fat marker genes (*Ucp1*, *Dio2*, *Ppara*, *Prdm16*, *Adrb3*, and *Vegfa2*; Fig. 2c) and, concomitantly, higher expression of markers of adipocyte functionality (*Pparg*, *Glut4*, *Fabp4*, *Adipoq*, and *Plin*; Fig. 2d) as well as of genes involved in oxidative metabolism (*Idh3a*, *Bckdhb*, *Cox7a1*, and *Ppargc1a*; Supplementary Fig. 1e) that clearly indicate transcriptional remodeling of IngWAT. Likewise, EpiWAT of H3atKO mice showed similar though milder phenotype, with no significant differences in adipocyte morphology compared to floxed mice (Supplementary Fig. 2d) but increased expression of browning genes, adipose markers, and oxidative metabolism genes (Supplementary Fig. 2e–g). Of note, levels of serum triglycerides (Supplementary Fig. 1f) and non-esterified FAs (NEFA) (Supplementary Fig. 1g) were reduced in H3atKO mice, while serum cholesterol levels did not differ (Supplementary Fig. 1h). Finally, livers from H3atKO, in which *Hdac3* was not ablated (Supplementary Fig. 1i), did not show signs of lipid steatosis, as shown by hematoxylin and eosin staining (Supplementary Fig. 1j).

The increased thermogenic activity during cold exposure in H3atKO mice prompted us to analyze BAT from floxed and H3atKO mice. Surprisingly, we found that *Hdac3* ablation in brown fat of mice housed at room temperature did not reprogram the expression of brown markers, except *Dio2*, nor that of genes involved in glucose and lipid metabolism (Supplementary Fig. 3c, f, g). More importantly, the expression of UCP1 protein did not increase in BAT of H3atKO mice (Supplementary Fig. 3d). HDAC3 is part of transcriptional repressor complexes and could potentially be involved in the maintenance of phenotypic features of different organs, including BAT. Thus, to assess whether *Hdac3* ablation unselectively de-repressed expression of genes typically not expressed in brown adipose tissue, we analyzed the expression of white fat selective markers (*Retn*, *Ednra*, *Serpina3k*, and *Psat1*) in BAT and found that none of these markers was increased in H3atKO mice (Supplementary Fig. 3e). In order to unveil the possible contribution of BAT to improved thermogenic function in H3atKO mice upon cold challenge, we analyzed gene expression profile in BAT of mice exposed to 4 °C for 24 h. We found no differences in the expression of brown markers and of most metabolic genes between floxed and H3atKO mice (Supplementary Fig. 3h). Furthermore, hematoxylin and eosin staining did not reveal morphological modifications between the two groups (Supplementary Fig. 3i). Altogether, these results indicate that *Hdac3* ablation selectively remodels white fat phenotype, revealing HDAC3 as an epigenetic regulator of WAT metabolism and function.

**HDAC3 ablation remodels white adipose tissue metabolome.** The profound differences of expression of metabolic genes in BAT and WAT (Fig. 1a) lead us to investigate whether browning occurring upon *Hdac3* ablation in WAT even in the absence of cold exposure determined a metabolic rewiring of this tissue. Metabolomic analysis of IngWAT unveiled a differential pattern of metabolites related to glycolysis and TCA cycle, FAs, and acyl carnitines in H3atKO vs. floxed mice (Fig. 3a). Volcano plot (Fig. 3b, Supplementary Table 1) showed that the metabolite mostly modulated upon *Hdac3* ablation was citrate, whose levels were reduced along with that of acetyl-CoA in IngWAT (Fig. 3c). To explain these metabolic changes, we measured the expression of genes involved in citrate metabolism: we detected increased expression of citrate synthase (*Cs*) in H3atKO mice (Fig. 3d), suggesting that oxaloacetate could condensate to acetyl-CoA to form citrate. Further gene expression analysis also demonstrated induction of the gene encoding the mitochondrial citrate carrier *Slc25a1*, which exports citrate from mitochondria to cytosol. The expression of the gene encoding for the ATP citrate lyase (*Acly*) cleaving citrate to acetyl-CoA and oxaloacetate (Fig. 3d) increased in H3atKO mice, explaining the reduction of citrate observed in these mice. Further metabolomic analysis also revealed reduced levels of glucose, glucose 6-phosphate, fructose 1,6-bisphosphate, and dihydroxyacetone phosphate/glyceraldehyde 3-phosphate in H3atKO (Fig. 3e), indicating reduced accumulation of glycolysis intermediates, potentially consistent with greater conversion of glucose to pyruvate. A similar gene expression profile was observed in EpiWAT from H3atKO mice (Supplementary Fig. 4a, b). In analogy to BAT (Fig. 1a), the expression of pyruvate kinase *Pkm* and *Pdk4* was highly augmented in IngWAT of H3atKO mice (Fig. 3f). Western blot analysis of pyruvate dehydrogenase (PDHE1α) showed increased phosphorylation at serine 293 and 300 (Fig. 3g) in IngWAT of H3atKO mice, correlating with reduced activity of the protein[20]. These results indicate that *Hdac3* deletion stimulates glycolysis, but at the same time prevents the conversion of pyruvate to acetyl-CoA. Metabolome analysis showed that both levels of pyruvate and lactate (Supplementary Fig. 5a) were reduced in WAT from H3atKO, excluding that in our model pyruvate is converted to lactate. In this regard, increased expression of *Mpc1* and *Pcx* in knock out mice (Fig. 3f) suggested that pyruvate entered into mitochondria, where it is converted to oxaloacetate. The expression of cytosolic PEPCK (*Pck1*) showed a robust increase in H3atKO mice (Fig. 3f). Of note, this enzyme has been proposed to play a relevant role in BAT metabolism and non-shivering thermogenesis[21]. We also found that intermediates of the pentose phosphate pathway were reduced in H3atKO mice (Supplementary Fig. 5b). Moreover, we found lower levels of TCA cycle intermediates in KO mice (Supplementary Fig. 5c), possibly as a consequence of considerable citrate export from mitochondria. These results point out that adipose ablation of *Hdac3* profoundly remodeled metabolome in WAT, most likely as a consequence of reset gene expression program.

---

**Fig. 4** *Hdac3* ablation activates a futile cycle of FA metabolism in adipocytes. **a** IngWAT fatty acid quantification by mass spectrometry in floxed and H3atKO mice ($n = 7$–8 per group); **b** Malonyl CoA levels in IngWAT of floxed and H3atKO mice ($n = 7$–8 per group); **c** Expression of lipolysis and FA β-oxidation genes in IngWAT of floxed and H3atKO mice ($n = 9$–11 per group); **d** Western blot analysis and quantification of ATGL and ACADL in IngWAT of floxed and H3atKO mice ($n = 3$ per group); **e** Expression of lipogenic genes in IngWAT of floxed and H3atKO mice ($n = 9$–11 per group); expression of *Acly* gene was the same reported in Fig. 3d; **f** Experimental paradigm of differentiation and infection of C3H/10T1/2 cells; **g** Western blot analysis of HDAC3, showing the deletion of the protein in cells infected with adenovirus expressing a shRNA targeted to *Hdac3* (shHDAC3) vs. cell infected with a scrambled control shRNA (scramble; $n = 3$); **h** Expression of browning genes in scramble and shHDAC3 infected cells ($n = 4$ biological replicates per group); **i** Morphology of scramble and shHDAC3 infected cells, scale bar is 50 μm.; **j** Expression of lipolytic and lipogenic genes in scramble and shHDAC3 infected cells ($n = 4$ biological replicates per group); **k** Glycerol release in scramble and shHDAC3 infected cells ($n = 10$ per group); **l–n** [13]C-labeled acetyl-CoA, citrate, and palmitate in scramble and shHDAC3 infected cells ($n = 6$ per group). Data are presented as mean ± s.e.m. Statistical analysis: Student's *t*-test, *$p < 0.05$, **$p < 0.01$, and ***$p < 0.001$

**Lack of HDAC3 elicits a futile cycle of FA metabolism in WAT**. A distinctive hallmark of BAT metabolism is the concomitant activation of lipolytic and lipogenic pathways to sustain energy demand related to thermogenic potential[17–19]. Interestingly, cold exposure activates a similar reprogramming of lipid metabolism in WAT via β3-adrenergic receptor[22]. Thus, we completed the metabolomics approach with lipidomics analysis in WAT of floxed and H3atKO mice, which revealed a reduction of saturated and unsaturated FAs in H3atKO (Fig. 4a) in the IngWAT. Decreased levels of FAs in adipose tissue may be suggestive of a higher utilization of fat as energy source in this depot. In this regard, mass spectrometry analysis showed that the allosteric inhibitor of CPT1 malonyl-CoA was reduced in IngWAT of H3atKO mice (Fig. 4b), thus favoring mitochondrial FA transport and β-oxidation. Accordingly, we detected an upsurge of the expression of lipolytic genes (*Lipe* and *Atgl*), peroxisomal FA β-oxidation (*Acox1*) and mitochondrial FA β-oxidation (*Acsl1, Cpt1b, Slc25a20, Acadl, Acadm*, and *Hadh*; Fig. 4c). Western blot analysis confirmed significant increased protein expression of adipose tissue triglyceride lipase (ATGL) and of long chain acyl-CoA dehydrogenase (ACADL; Fig. 4d). The expression of genes involved in de novo FA synthesis (*ChREBPβ, Srebp1c, Acly, Fasn, Acaca, G6pd2*, and *Me3*) was upregulated in H3atKO mice (Fig. 4e), corroborating the hypothesis that the browning effect observed in these mice is sustained by a metabolic rewiring that phenocopies BAT metabolism. Similar, albeit less pronounced, effects in gene expression and FA profile occurred in epididymal fat (Supplementary Fig. 4c–e). As expected, cold exposure increased the expression of browning markers in IngWAT from mice exposed to 4 °C for 24 h (Supplementary Fig. 6), and similarly to *Hdac3* ablation, increased expression of glucose-related genes (*Glut4* and *Pdk4*) and of genes involved in citrate export and FA synthesis/β-oxidation (Supplementary Fig. 6). Interestingly, H3atKO mice exposed to cold showed less marked effects when compared to floxed under the same temperature condition (Supplementary Fig. 6).

It has been demonstrated that the high rate of FA β-oxidation correlates with increased levels of short-chain acylcarnitines[23]. Nonetheless, mass spectrometry analysis yielded unexpected results as the levels of most acylcarnitines in both inguinal (Supplementary Fig. 7a) and epididymal WAT (Supplementary Fig. 7b) showed a reduction in H3atKO mice. To explain the decreased levels of acylcarnitines in H3atKO mice we hypothesized that FAs are shortened via peroxisomal β-oxidation and that acetyl-CoA from FA β-oxidation in adipose tissue of H3atKO mice is immediately used for de novo FA synthesis. Consistent with increased peroxisomal FA β-oxidation we also detected increased expression of Acox1 (Fig. 4c) and Catalase, the latter involved in detoxification of $H_2O_2$ resulting from peroxisomal β-oxidation. To demonstrate the onset of a futile cycle of FA metabolism consequent to HDAC3 inactivation, we used a cellular model of preadipocytes, C3H/10T1/2. Preadipocytes were infected with scramble adenovirus or with adenovirus delivering short hairpin RNA to silence Hdac3 (shHDAC3; Fig. 4f). Silencing of HDAC3 in cells was assessed by western blot (Fig. 4g). This cellular model recapitulated the *in vivo* effects of *Hdac3* deletion: *in vitro* inactivation of Hdac3 caused the induction of browning genes (Fig. 4h), and promoted adipocyte differentiation, as demonstrated by the greater number of differentiated cells (Fig. 4i) and by the increased lipid accumulation in ORO staining (Supplementary Fig. 8a). Accordingly, shHDAC3 treated cells showed increased expression of adipocyte markers (Supplementary Fig. 8b). Gene expression analysis also confirmed that silencing of *Hdac3* in C3H/10T1/2 adipocytes increased the expression of genes of glucose and citrate metabolism (Supplementary Fig. 8c) and of lipolysis, FA β-

oxidation and FA synthesis genes (Fig. 4j). The increased rate of lipolysis was confirmed by higher glycerol release in shHDAC3 cells (Fig. 4k). Administration of [U-$^{13}$C]Palmitate (where U denotes uniformly labeled) demonstrated that silencing of *Hdac3* determined higher β-oxidation rate, as confirmed by increase of $^{13}$C-Acetyl-CoA (+1) (Fig. 4l), coming from β-oxidation of labeled palmitate (Supplementary Fig. 9). No differences in the levels of $^{13}$C-Acetyl-CoA (+2) between scramble and shHDCA3-silenced cells were detected (Fig. 4l). Higher levels of $^{13}$C-Acetyl-CoA (+1) were paralleled by decreased levels of $^{13}$C-Citrate (+1) (Fig. 4m), confirming that silencing of HDAC3 promoted the activity of ATP citrate lyase, which converts citrate into acetyl-CoA. This result was in line with the upregulation of the gene (*Acly*) encoding for this enzyme (Fig. 4j). $^{13}$C-isotopologue labeling experiments also demonstrated that shHDAC3 silenced cells showed higher levels of $^{13}$C-Citrate (+4) (Fig. 4m), whose formation requires that 2 molecules of $^{13}$C-Acetyl-CoA (+2), coming from labeled palmitate, entered the TCA cycle (Supplementary Fig. 9). No differences in the levels of $^{13}$C-Citrate (+2) and $^{13}$C-Citrate (+5) between scramble and shHDCA3-silenced cells were detected (Fig. 4m). Increased $^{13}$C-Acetyl-CoA (+1) and $^{13}$C-Citrate (+4) levels confirmed that silencing of *Hdac3* boosted mitochondrial FA β-oxidation and TCA cycle. Moreover, we found higher level of $^{13}$C-Palmitate (+2) in shHDAC3 cells (Fig. 4n and Supplementary Fig. 9) and a trend of increased of $^{13}$C-Malonyl-CoA (+2, +3) (Supplementary Fig. 8d), corroborating de novo FA synthesis from β-oxidized FAs under this condition. $^{13}$C-Acetyl-CoA (+2) is incorporated in $^{13}$C-Palmitate (+2), thus explaining the lack of increase of this isotopologue in shHDAC3 cells. These *in vitro* experiments demonstrated that adipose ablation of HDAC3 triggers FA futile cycle in white fat.

**HDAC3 deletion selectively remodels histone acetylation**. To investigate the molecular mechanisms underlying the effect of HDAC3 ablation in white adipocytes, we focused on histone post-translational modifications as read out of HDAC3 activity. First we tested whether HDAC3 deletion correlated with increased histone acetylation. Consistent with evidences reported by other authors[7, 24], acidic extraction of histones resulted in no differences in global pan- and lysine 27 acetylation on histone H3 in mice and cells lacking HDAC3 (Supplementary Fig. 10a, b). Additionally, in order to investigate whether acetyl-CoA coming from FA β-oxidation could be a source of acetyl groups for histone acetylation, we performed a mass spectrometry analysis of histone H3 (acid extracted) from C3H10T1/2 cells infected with scramble or HDAC3 short hairpin RNAs (shRNAs), incubated with [U-$^{13}$C]-Palmitate. Consistent with the western blots, mass spectrometry analysis showed no statistical significant differences in acetylation status of 9–17, 18–26, 27–40, 54–63, 73–83, and 117–128 peptides (Supplementary Fig. 10c). More interestingly, we were able to show incorporation of labeled carbons ($^{13}$C acetylation) on 9–17, 18–26, and 27–40 peptides, with no differences between scrambled and knock down cells. Peptide 27–40 contains 3 lysine residues that could be acetylated (Lys 27, 36, and 37). Therefore, we specifically investigated the acetylation status of lysine 27, a known mark of active enhancers, and quantified the amount of $^{12}$C- vs. $^{13}$C-acetylation in scrambled or HDAC3 silenced cells (Supplementary Fig. 10d). Since we detected that most acetyl groups showed labeled carbons ($^{13}$C-acetylation) in both conditions, we confirmed that acetyl-CoA, derived from FA β-oxidation, was a source of acetyl groups for histone acetylation on lysine 27. This result corroborated data by $^{13}$C-isotopologue labeling experiments (Fig. 4l–n): knock down of HDAC3 in fact increased β-oxidation and amount of labeled acetyl-CoA, but concomitantly this was in part used for

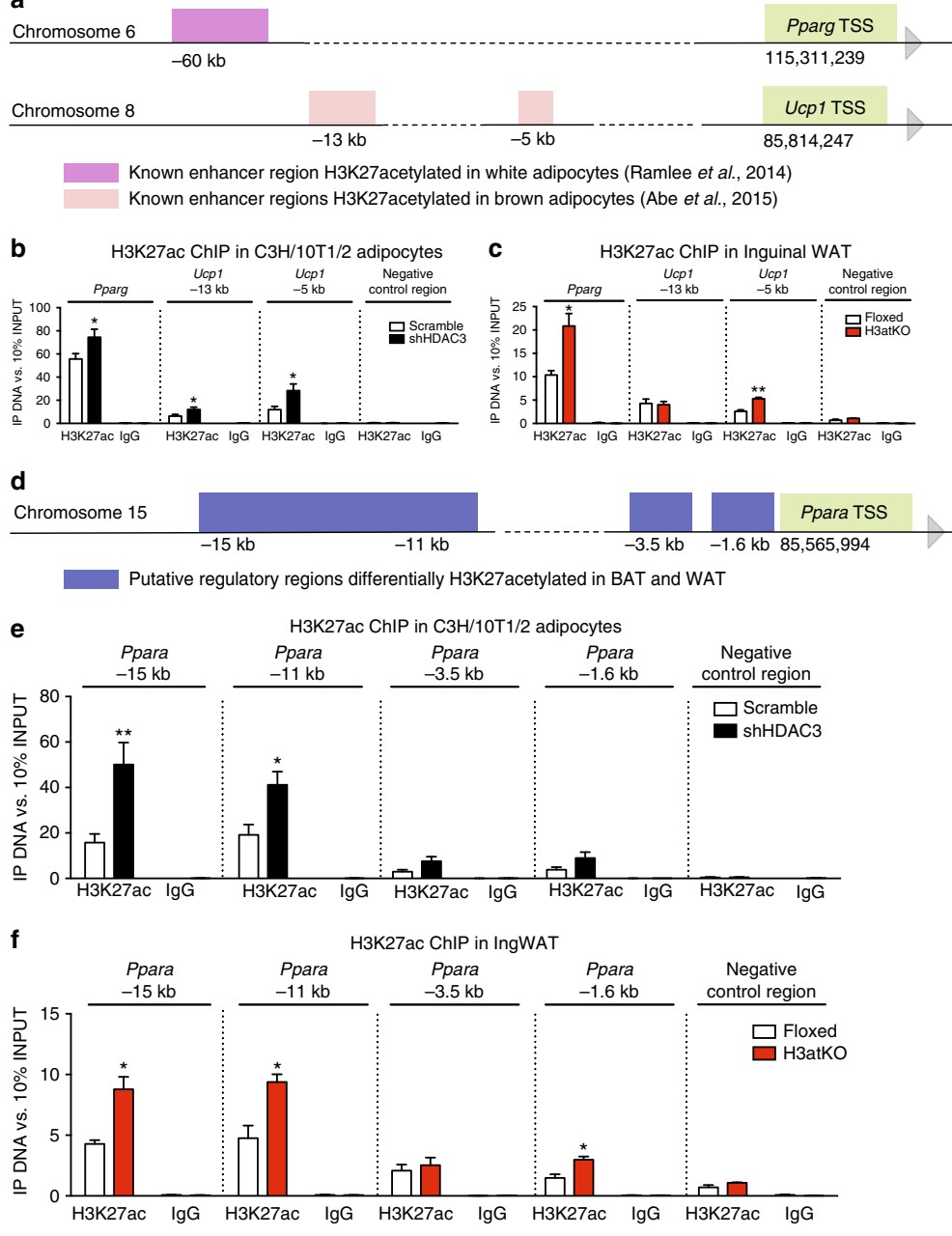

**Fig. 5** Adipose ablation of HDAC3 remodels chromatin acetylation in "hot regions". **a** Schematic representation of known *Pparg* and *Ucp1* enhancers; **b** ChIP analysis showing H3K27ac enrichment on *Pparg* and *Ucp1* enhancers in C3H/10T1/2 cells with *Hdac3* knock down ($n = 6$ per group); **c** ChIP analysis showing H3K27ac enrichment on *Pparg* and *Ucp1* enhancers in IngWAT of H3atKO mice ($n = 3$ per group, where each sample represent a pool from two individual mice); **d** Schematic representation of putative regulatory regions of *Ppara* gene; **e** ChIP analysis showing H3K27ac enrichment on putative regulatory regions of *Ppara* gene in C3H/10T1/2 cells with *Hdac3* knock down ($n = 6$ per group) **f** ChIP analysis showing H3K27ac enrichment on putative regulatory regions of *Ppara* gene in IngWAT of H3atKO mice ($n = 3$ per group where each sample represent a pool from 2 individual mice). Data are presented as mean ± s.e.m. Statistical analysis: Student's *t*-test, *$p < 0.05$ and **$p < 0.01$

de novo synthesis of FA. This possibly caused the lack of increased $^{13}$C acetyl-CoA incorporation on histone H3.

It has been reported that selective deletion of HDAC3 increased acetylation of specific chromatin regions, even though global histone acetylation did not change[7]. Thus, we focused on "hot regions", whose epigenomic regulation might be crucial for the transcriptional and metabolic phenotype of H3atKO mice. Since Hdac3 deletion increased expression of the master regulator of adipocyte differentiation, *Pparg*, and of its main targets, we looked at acetylation status of this gene. It has been recently demonstrated that H3K27ac marks a potent enhancer of *Pparg*

gene[25] (Fig. 5a), thus we performed a ChIP analysis of H3K27ac and found increased acetylation on *Pparg* enhancer in shHDAC3 treated cells and in IngWAT of H3atKO mice (Figs. 5b, c). Another key gene upregulated upon *Hdac3* inactivation is *Ucp1*: two enhancers located 13 and 5 kb upstream of the transcription start site (TSS) of *Ucp1* are hyperacetylated (H3K27ac) in differentiating brown adipocytes[15] (Fig. 5a). ChIP analysis showed increased H3K27 acetylation in cells lacking HDAC3 (Fig. 5b), while only −5kb enhancer resulted in hyperacetylated in IngWAT of H3atKO mice (Fig. 5c). Moreover, by reviewing published H3K27ac ChIP-Seq data (GSE63964), we

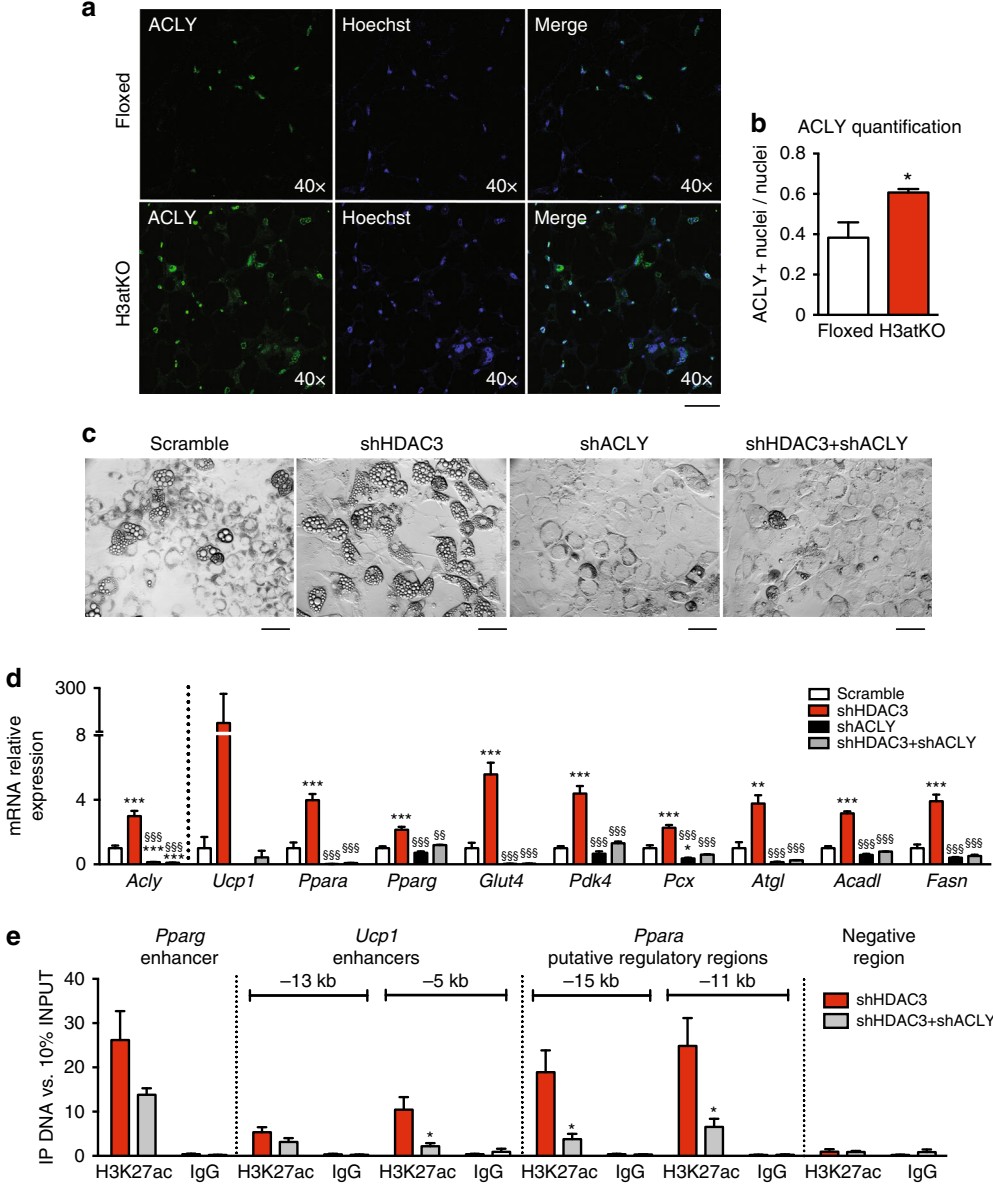

**Fig. 6** ACLY is required for the establishment of H3atKO phenotype. **a**, **b** Immunofluorescence staining and quantification of ACLY, showing nuclear localization in *Hdac3* knock out mice fed LFD ($n = 3$ per group), scale bar is 40 μm; **c** Morphology of cells infected with adenovirus expressing shRNA targeted to *Hdac3* (shHDAC3), to *Acly* (shACLY) or both (shHDAC3 + shACLY) or to a scrambled control shRNA (scramble), scale bar is 50 μm; **d** Gene expression analysis in knock down cells ($n = 3$ per group); **e** H3K27ac ChIP analysis in knock down cells ($n = 6$ per group). Data are presented as mean ± SEM. Statistical analysis: One way ANOVA, Tukey as post hoc test, *$p < 0.05$, **$p < 0.01$, ***$p < 0.001$ vs. scramble, §§$p < 0.01$, §§§$p < 0.001$ vs. shHDAC3

noticed hyperacetylation of H3K27 in regions located 15, 11, 3.5, and 1.6 kb upstream of the TSS of *Ppara* gene in BAT samples, while the same regions were not acetylated in WAT samples (Fig. 5d, Supplementary Fig. 11a). PPARα is in fact highly expressed in BAT and directly regulates *Pdk4* and FA β-oxidation genes[26]. Thus, we analyzed H3K27 acetylation in these putative regulatory regions and found that −15 and −11 kb regions showed significant increase in H3K27ac in shHDAC3 cells and in IngWAT of mice lacking HDAC3 (Fig. 5e, f). Moreover, H3K27 acetylation increased in the −1.6 kb region in IngWAT of H3atKO mice (Fig. 5f). Interestingly, H3K27ac ChIP analysis in BAT from floxed and H3atKO mice showed no differences in the acetylation of *Pparg* and *Ucp1* enhancers (Supplementary Fig. 11b), and of putative regulatory regions of *Ppara* (Supplementary Fig. 11c), confirming that HDAC3 does

not play a role in determining the phenotype of BAT, as opposed to WAT.

**ACLY is required to establish the phenotype of H3atKO mice.** In WAT from H3atKO mice we detected increased expression of *Acly* gene (Fig. 4e), encoding the cytosolic enzyme belonging to the FA biosynthetic pathway that converts citrate to acetyl-CoA and oxaloacetate[27]. However, the reaction catalyzed by ACLY also represents the major source of acetyl-CoA for histone acetylation in mammalian cells[13]. Notably, immunofluorescence analysis of ACLY in IngWAT from floxed and H3atKO mice demonstrated that this enzyme localized also in the nucleus, and most strikingly we detected higher nuclear localization in knock out mice (Fig. 6b). Nuclear ACLY catalyzes the reaction that could locally provide acetyl groups to sustain chromatin hyperacetylation

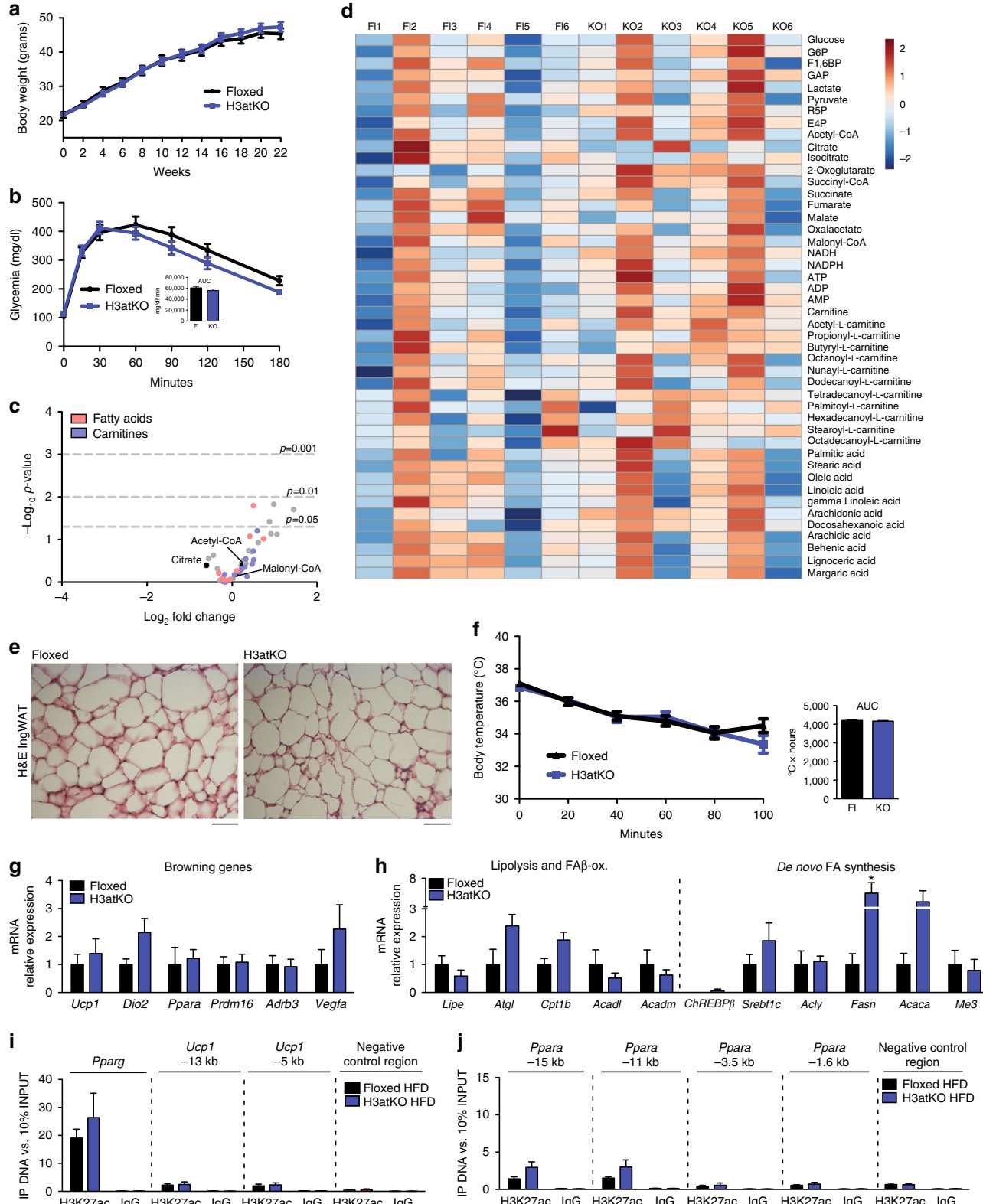

**Fig. 7** High fat diet overcomes the effects of adipose ablation of HDAC3. **a** Body weight gain of floxed and H3atKO mice fed HFD ($n = 12$–14 per group); **b** Glucose tolerance test of floxed and H3atKO mice ($n = 12$–14 per group), performed after 20 weeks of HFD; **c** Volcano plot of IngWAT metabolome in floxed and H3atKO mice fed HFD ($n = 6$ per group); **d** Heatmap of IngWAT metabolome in floxed and H3atKO mice fed HFD ($n = 6$ per group); **e** Hematoxylin–eosin staining of IngWAT from floxed and H3atKO mice fed HFD, scale bar is 100 μm; **f** Body temperature of HFD floxed and H3atKO mice exposed to 4 °C for 100 min ($n = 8$ per group); **g, h** Gene expression analysis of browning genes and lipolytic/lipogenic genes in IngWAT of floxed and H3atKO mice ($n = 5$ per group); **i, j** H3K27ac ChIP analysis showing lack of enrichment on *Pparg* and *Ucp1* enhancers and on putative regulatory regions of *Ppara* gene in IngWAT from H3atKO HFD mice ($n = 3$ per group). Data are presented as mean ± s.e.m. Statistical analysis: Student's *t*-test, *$p < 0.05$

observed in *Pparg* and *Ucp1* enhancers and in *Ppara* putative regulatory regions. To demonstrate that acetylation of these regions in adipocytes lacking HDAC3 was dependent on the increased acetyl-CoA arising from ACLY-mediated cleavage of citrate, we co-silenced both ACLY and HDAC3 in C3H/10T1/2 cells with shRNAs. Efficiency of ACLY knock down was confirmed by western blot (Supplementary Fig. 12a) and by qPCR (Fig. 6d). Interestingly, we found that ACLY silencing completely blocked adipogenesis as shown by dramatically decreased number of differentiated adipocytes (Fig. 6c, third panel) and by reduced

expression of adipose markers *Pparg* and *Glut4* (Fig. 6d, black bars). Most importantly, co-silencing of ACLY and HDAC3 completely abrogated the effects of HDAC3 inactivation, since few cells appeared fully differentiated to adipocytes (Fig. 6c, fourth panel). Expression analysis showed that genes that were strongly induced upon HDAC3 silencing (*Ucp1*, *Ppara*, *Pparg*, *Glut4*, *Pdk4*, *Pcx*, *Atgl*, *Acadl*, and *Fasn*), were significantly reduced in cells lacking ACLY as well as in cells lacking both HDAC3 and ACLY (Fig. 6d). ChIP analysis showed significant reduction of H3K27 acetylation on −5 kb *Ucp1* enhancer and on

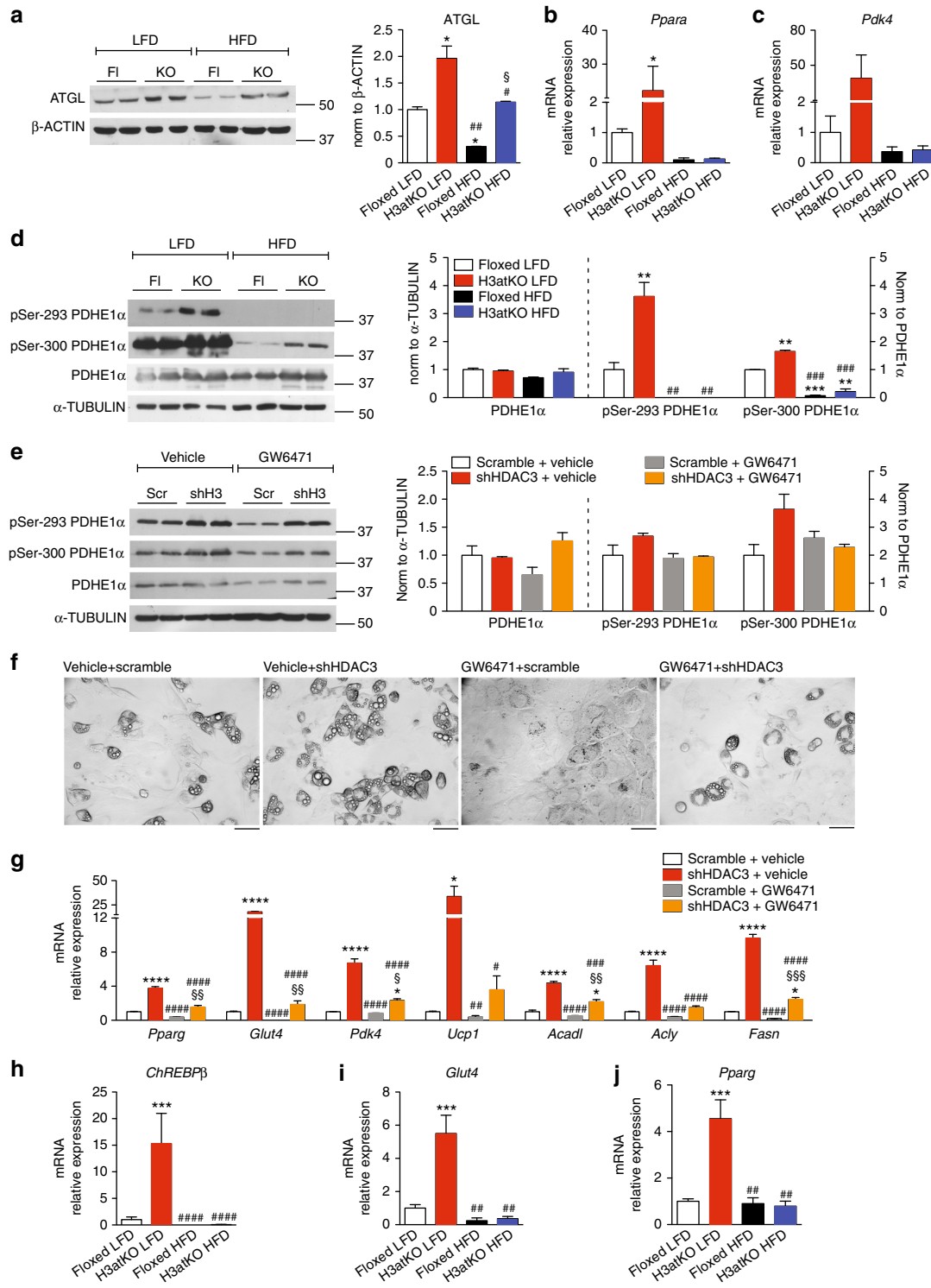

*Ppara* putative regions in co-silenced cells compared to shHDAC3 treated cells, while a 50% reduction, though not statistically significant, was observed for *Pparg* enhancer (Fig. 6e). Since these results clearly demonstrate the pivotal role of ACLY in determining the phenotype of HDAC3 knocked down cells, we infer that ACLY is a central player in the transcriptional and metabolic outcome of *Hdac3* ablation in mice. Another potential source of acetyl groups for de novo FA synthesis is acetate that can be converted to acetyl-CoA by cytosolic acetyl-CoA synthetase (ACSS2). Interestingly, *Acss2* gene was strongly induced in IngWAT of H3atKO mice (Supplementary Fig. 12b). However, knock down of ACSS2 (Supplementary Fig. 12c) did not affect adipocyte morphology (Supplementary Fig. 12d) and gene expression (Supplementary Fig. 12e) when compared to scramble shRNA. Also, as opposed to ACLY shRNA, co-silencing of ACSS2 and HDAC3 by shRNA did not result in any morphological or transcriptional (Supplementary Fig. 12d, e) effect as compared to cells treated with shHDAC3. Our results mirror the observations of Wellen et al.[13] who showed that ACLY is the key player in coordinating nuclear activity (that is, chromatin acetylation) and cellular metabolism, whereas ACSS2, the other enzyme potentially providing extra mitochondrial acetyl-CoA, plays a minor or an ancillary role in these processes.

**No effect of *Hdac3* KO on lipid metabolism with high fat diet.** Having observed profound remodeling of WAT metabolism in H3atKO mice, we investigated whether adipose ablation of HDAC3 was able to prevent or attenuate diet-induced obesity. To this end, *Hdac3* floxed and knock out mice were fed for 24 weeks with an obesogenic high fat diet (HFD). Intriguingly, we found no differences in body weight gain (Fig. 7a) and in glucose tolerance (Fig. 7b). Metabolomic analysis (Fig. 7c, d, Supplementary Table 2) of IngWAT showed that effects on glycolysis, TCA cycle metabolites (Supplementary Fig. 13a–c), intermediates of the pentose phosphate pathway (Supplementary Fig. 13d), FAs (Supplementary Fig. 13e), and acyl carnitines (Supplementary Fig. 13f), observed in H3atKO fed LFD, were lost in mice fed HFD. Accordingly, expression analysis of genes related to glucose and citrate metabolism (Supplementary Fig. 14a, b) showed no change. Genes involved in lipolysis and FA β-oxidation were not significantly upregulated in knock out mice (Fig. 7h, Supplementary Fig. 14c), while among genes of FA synthesis only *Fasn* was significantly increased, though to a lesser extent (6-fold increase in H3atKO, Fig. 7h) as compared to mice fed low fat diet (LFD, 43-fold increase in H3atKO, Fig. 4e). Moreover, no morphological differences in inguinal adipocytes from floxed and H3atKO mice were highlighted by histological analysis (Fig. 7e). As a consequence of the lack of metabolic remodeling, IngWAT of H3atKO mice fed HFD did not undergo browning, as cold

challenge experiment showed no differences in maintenance of body temperature in floxed and H3atKO mice exposed to 4 °C (Fig. 7f), and browning genes were not upregulated in knock out mice (Fig. 7g). Also, upregulation of adipocyte functionality genes (including *Pparg*) and of oxidative metabolism genes was lost upon HFD feeding (Supplementary Fig. 14c). No difference in global pan- and lysine 27 acetylation of histone H3 was detected upon *Hdac3* ablation (Supplementary Fig. 15a, b). Moreover, we found low expression of *Acly* in HFD fed mice, and no upregulation of this gene in response to *Hdac3* ablation (Supplementary Fig. 15c). This result was paralleled by immunofluorescence analysis, showing no differences in the number of ACLY positive nuclei in IngWAT of floxed and knock out mice (Supplementary Fig. 15d, e), and a statistically significant reduction of ACLY positive nuclei compared to floxed and H3atKO fed LFD (Supplementary Fig. 15d, e). Accordingly, ChIP analysis demonstrated that ablation of *Hdac3* did not affect H3K27 acetylation of *Pparg* and *Ucp1* enhancers, and of *Ppara* putative regulatory regions in presence of a high fat feeding (Figs. 7i, j).

As shown above, the peculiar phenotype observed in H3atKO is sustained by rewiring of lipid metabolism, with concomitant activation of FA synthesis and β-oxidation genetic networks resulting in the onset of a futile cycle. To gain insight on how HFD overcomes the metabolic and transcriptional effects of adipose ablation of *Hdac3*, we focused on lipid metabolic pathways that can be modulated in adipose tissue upon high fat feeding. It has been demonstrated that HFD reduces lipolysis mediated by cathecolamines in adipose tissue, and that high fat feeding reduces levels of lipases[28]. Among these, particularly relevant is ATGL, since its activity is fundamental to feed the futile cycle of FAs in adipose tissues during β3-adrenergic activation and to maintain brown-like phenotype[22]. We measured ATGL expression in IngWAT of floxed and H3atKO mice fed LFD or HFD and we found that ablation of *Hdac3* increased expression of ATGL with both diets. However, when fed HFD floxed and knock out mice showed significantly lower levels of ATGL as compared to H3atKO LFD fed mice (Fig. 8a). This observation suggests that the lipolytic activity of IngWAT releasing free FAs (FFAs) is reduced upon HFD feeding. Notably, knock down of ATGL in brown adipocytes prevented induction of genes involved in FA β-oxidation, owing to lack of lipolytic products (FFAs) that act as ligands for PPARs[29]. In our model a key gene regulated upon *Hdac3* ablation in IngWAT is *Pdk4*, whose protein product phosphorylates and inactivates pyruvate dehydrogenase (PDHE1α), thus allowing the entrance of oxaloacetate in the TCA cycle to form citrate. Importantly, *Pdk4* is transcriptionally activated by PPARα[7]. As previously shown (Fig. 7j), H3K27 hyperacetylation on *Ppara* putative

**Fig. 8** Role of PPARα and ChREBPβ in lipolysis and lipogenesis in WAT and in adipocytes. **a** Western blot analysis and quantification of ATGL in IngWAT from floxed and H3atKO mice fed LFD or HFD ($n = 2$, where each replicate represent a pool of samples from 3 mice); **b**, **c** mRNA expression of *Ppara* and *Pdk4* in IngWAT from floxed and H3atKO mice fed LFD or HFD ($n = 9$–11 for LFD mice, $n = 5$ per group for HFD samples); **d** Western blot and quantification of pSer-293 PDHE1α, pSer-300 PDHE1α, PDHE1α in IngWAT of *Hdac3* floxed and knock out mice fed LFD or HFD ($n = 2$, where each replicate represent a pool of samples from 3 mice); Data are presented as mean ± s.e.m. Statistical analysis: two way ANOVA, Tukey as *post hoc* test, *$p < 0.05$, **$p < 0.01$, ***$p < 0.001$ vs. Floxed LFD, #$p < 0.05$, ##$p < 0.01$, ###$p < 0.001$ vs. H3atKO LFD, §$p < 0.05$, §§$p < 0.01$, §§§$p < 0.001$ vs. Floxed HFD. **e** Western blot and quantification of pSer-293 PDHE1α, pSer-300 PDHE1α, PDHE1α in cells infected with adenovirus expressing shRNA targeted to *Hdac3* or to a scrambled control shRNA (scramble) and differentiated in presence of vehicle or 10 μM GW6471 ($n = 2$, where each replicate represent a pool of 3 samples); **f** Morphology of cells infected with adenovirus expressing shRNA targeted to *Hdac3* or to a scrambled control shRNA (scramble) and differentiated in presence of vehicle or 10 μM GW6471, scale bar is 50 μm; **g** Gene expression analysis on mRNA from cells infected with adenovirus expressing shRNA targeted to *Hdac3* or to a scrambled control shRNA (scramble) and differentiated in presence of vehicle or 10 μM GW6471 ($n = 3$ per group). Data are presented as mean ± s.e.m. Statistical analysis: two-way ANOVA, Tukey as *post hoc* test, *$p < 0.05$, **$p < 0.01$, and ***$p < 0.001$ vs. Scramble + Vehicle, #$p < 0.05$, ##$p < 0.01$, and ###$p < 0.001$ vs. shHDAC3 + Vehicle, §$p < 0.05$, §§$p < 0.01$, and §§§$p < 0.001$ vs. Scramble + GW6471. **h–j** mRNA expression of ChREBPβ, Glut4, Pparg in IngWAT of *Hdac3* floxed and knock out mice fed LFD or HFD ($n = 9$–11 for LFD mice, $n = 5$ per group for HFD samples); Two-way ANOVA, Tukey as *post hoc* test, *$p < 0.05$, **$p < 0.01$, and ***$p < 0.001$ vs. Floxed LFD, #$p < 0.05$, ##$p < 0.01$, and ###$p < 0.001$ vs. H3atKO LFD

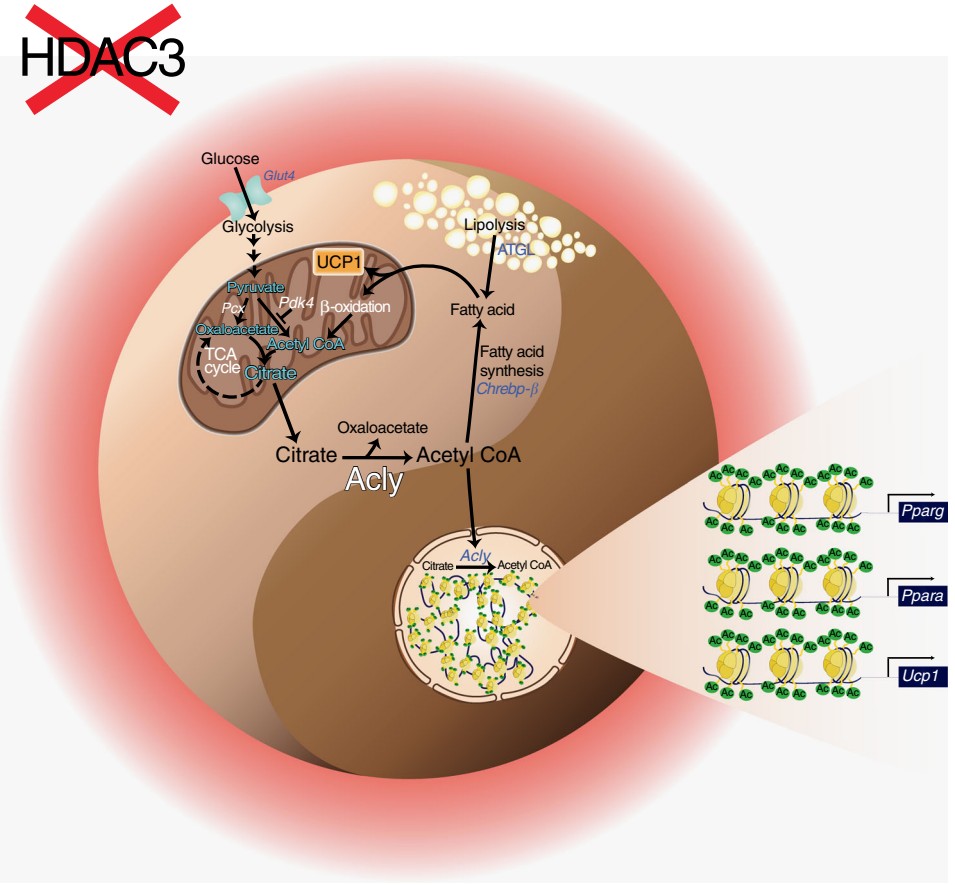

**Fig. 9** Transcriptional and metabolic switch occurring in WAT of *Hdac3* KO mice. Metabolism and epigenetic remodeling are complementary and mutually interconnected as each of these phenomena could influence the other, giving rise to a specific changes in cellular physiology. In our model, genetic ablation of the epigenetic modifier HDAC3 in WAT imposes a futile cycle of fatty acid utilization and synthesis that provides acetyl-CoA to remodel specific chromatin regions, ultimately determining WAT browning. High-fat diet feeding perturbs this subtle equilibrium and prevents the epigenome remodeling required for rewiring metabolism in WAT

regulatory regions in response to *Hdac3* ablation was blunted in IngWAT of HFD fed mice. Consequently, *Ppara* expression was significantly reduced in floxed and H3atKO mice fed HFD when compared to H3atKO mice fed LFD (Fig. 8b). Moreover, HFD feeding, by blocking ATGL reduced the availability of FFAs for PPARα activation. As a result of lower transcription and activation of PPARα, we detected lower expression of *Pdk4* in floxed and H3atKO mice fed HFD when compared to H3atKO mice fed LFD (Fig. 8c). Moreover, HFD reduces phosphorylation of PDHE1α at Ser293[30]. In fact, both Ser293 and Ser300 resulted significantly less phosphorylated in floxed and H3atKO mice fed HFD (Fig. 8d), suggesting that the PDHE1α kinase activity of PDK4 in WAT from these mice was reduced. To elucidate the role of PPARα as mediator of effects of metabolic remodeling induced by *Hdac3* ablation, cells infected with scramble or HDAC3 shRNAs were treated with vehicle or with the PPARα antagonist GW6471, to mimic *in vitro* the lack of activation of PPARα that occurs in HFD regimen. *Hdac3* knock down in C3H/10T1/2 adipocytes tended to increase phosphorylation status of PDHE1α, while no increase was found upon treatment with GW6471 (Fig. 8e). Consistent with published literature[31], treatment with PPARα antagonist decreased the rate of differentiation (Fig. 8f), reducing expression of adipocyte markers *Pparg* and *Glut4* (Fig. 8g). Also, the expression of *Pdk4* and other key genes upregulated by shHDAC3 (*Ucp1*, *Acadl*, *Acly*, and *Fasn*) was significantly reduced in presence of GW6471 (Fig. 8g), implying that ablation of *Hdac3* induced expression of these genes, at least in part, via PPARα.

Another key point in the metabolic remodeling in WAT of H3atKO mice was the strong activation of a lipogenic program, that leads to higher level of ACLY, allowing a concomitant chromatin rearrangement that leads to increased expression of genes involved in higher metabolic requirements of adipocytes. As shown above, de novo FA synthesis as well as chromatin remodeling was abrogated in H3atKO mice fed HFD. It has been elegantly demonstrated that adipose tissue lipogenesis is primarily regulated by ChREBP, and that, among ChREBP isoforms, the most affected by HFD is ChREBPβ isoform[32]. Interestingly, the expression of this gene was significantly increased in H3atKO mice fed LFD, but, consistent with published literature, we found reduction of *ChREBPβ* upon HFD feeding in both floxed and knock out mice (Fig. 8h). Analyzing published H3K27ac ChIP-Seq data[33] (GSE63964), we noticed several regions located upstream and downstream the TSS for Exon1b of *ChREBP* gene (the alternative first exon for *ChREBPβ* isoform) showing higher H3K27 acetylation profile in BAT vs. WAT samples (Supplementary Fig. 16a, b). However, H3K27ac ChIP analysis showed no increase in acetylation status of these putative regulatory regions in IngWAT of H3atKO compared to floxed mice fed LFD (Supplementary Fig. 16c), ruling out that ablation of HDAC3 drives *ChREBPβ* expression by directly modulating histone acetylation levels. Importantly, it is known that *ChREBPβ* gene transcription is regulated in response to glucose levels through ChREBPα and it has been demonstrated that increased expression of the glucose transporter *Glut4* correlates with higher *ChREBPβ* expression[32]. *Glut4* in adipose tissue is

transcriptionally regulated by PPARγ[34]. Accordingly, we found strong upregulation of *Glut4* and *Pparg* genes in H3atKO mice fed LFD, while the expression of these genes was reduced in mice fed HFD (Fig. 8i, j). These results mirror ChIP and gene expression analyses, showing significant increase of *Pparg* enhancer H3K27 acetylation and of *Pparg* expression only in H3atKO mice fed LFD (Figs. 5c and 7j).

## Discussion

In this paper we disentangled the role of the histone modifier HDAC3 in adipose tissue physiology. The main finding we report is that adipose specific ablation of *Hdac3* affects FA metabolism and induces browning of white fat, promoting the global thermogenic capacity, without affecting BAT phenotype. Transcriptome analysis of IngWAT proved that ablation of HDAC3 did not de-repress globally gene transcription, demonstrating that the lack of HDAC3 does not lead to general reactivation of silenced genes. We demonstrated that remodeling of WAT phenotype is sustained by the metabolic rewiring of glucose and lipid metabolism (Fig. 9). Our results highlight the anaplerotic function of glucose/glycolysis to provide oxaloacetate to TCA cycle in our knock out model. In this frame, FAs represent an alternative source of acetyl-CoA thus explaining the exceptional activation of FA lipolysis and β-oxidation in WAT of mice lacking HDAC3. Concomitantly, mitochondrial citrate is exported to cytosol, where it is required for de novo lipogenesis. Thus, by exploiting this animal model, we learnt that WAT browning induced by HDAC3 ablation activates a "futile cycle" to meet the high metabolic demand of brown-like fat, whereby FFAs activate UCP1 and their oxidation provides reduced coenzymes (that is, NADH and FADH$_2$) required to electron transport chain. Such futile cycle seems to be operative in tissues with high-energy expenditure, such as BAT[17, 18, 22] as a way to cope with the need of great amount of energy.

Other authors described a different futile cycle, operating in brown and beige adipocytes, involving creatine metabolism to sustain adipose tissue energy expenditure and thermogenic capacity, demonstrating that creatine enhances respiration in beige-fat mitochondria when ADP is limiting[35]. These evidences, along with our experiments, suggest that beige/brown adipocytes can trigger paradoxical metabolic pathways to sustain the incredibly high metabolic demand during thermogenic activation.

We are tempted to speculate that such futile cycle of FA metabolism could be operative in other tissues in conditions of high energy requirements, such as skeletal muscle during sustained physical exercise, or heart. Future investigations will be required to test this fascinating hypothesis and to verify whether HDAC3 is involved in the regulation of the metabolic phenotype in these organs.

Intriguingly, WAT from mice exposed to cold (4 °C) showed similar profile of *Pdk4* expression as well as of genes of FA futile cycle. Based on these results it is tempting to speculate that *Hdac3* ablation phenocopies, at least in part, the transcriptional program elicited by cold exposure. Future experiments will focus on whether and how HDAC3 is involved in the adaptive response to cold exposure.

PDK4 and ACLY, which are upregulated upon *Hdac3* ablation, are key actors in establishing the phenotype of *Hdac3* knock out mice. PDK4 blocks the conversion of pyruvate to acetyl-CoA, thus making FA β-oxidation the alternative source of acetyl-CoA. On the other hand, ACLY is a key enzyme in lipogenesis and also provides acetyl-CoA for histone acetylation. Selective hyperacetylation of *Ucp1* and *Pparg* enhancers and *Ppara* putative regulatory regions in adipocytes lacking HDAC3 are lost when ACLY is also knocked down concomitantly with HDAC3,

highlighting the crucial role of this lipogenic enzyme in establishing the phenotype induced upon HDAC3 ablation. Our results demonstrate that HDAC3 directly regulates *Ucp1* transcription, but it also plays a role in the regulation of PPARα and PPARγ, which are known to drive UCP1 expression[36].

PPARα stimulates FA β-oxidation, but it is also essential in switching off the flow of glucose to acetyl-CoA, which is channeled to oxaloacetate, by increasing PDK4 expression. PPARα mediates, at in least in part, the effects of *Hdac3* knock down, as confirmed in the experiment with the antagonist GW6471. On the other hand, PPARγ may be important for FA uptake and usage by promoting triglycerides hydrolysis via LPL and ATGL (PPARγ target genes) that provide FAs as fuel available for high energy expenditure. Furthermore, PPARγ regulates *Glut4* expression[34], favoring glucose internalization and expression of ChREBPβ, one of the main regulators of lipogenesis in adipose tissue[32]. Thus, PPARα and γ emerged as key targets of HDAC3, whose de-repression upon *Hdac3* ablation induces WAT browning. A key question is how HFD completely offsets the effects of HDAC3 genetic inactivation. It should be noted that PPARs are activated by FFAs, mainly resulting from lipolytic action of ATGL. It has been shown that high fat feeding reduces protein expression of ATGL[28], therefore lower rate of lipolysis and reduced availability of FFAs are expected to strongly blunt PPAR activation in both floxed and *Hdac3* knock out mice fed HFD. The lack of PPARα activation determines lower *Pdk4* transcription, resulting in loss of pyruvate dehydrogenase phosphorylation in HFD fed mice, that allows a different glucose utilization and reduces the rate of FA β-oxidation. Intriguingly, high fat feeding also reduced expression of *Pparg* and *Ppara* genes, further reducing the transcription of their target genes. These effects, paralleled by lower expression of *ChREBPβ* and of its lipogenic target genes in response to HFD, disrupt the epigenetic and metabolic remodeling induced by adipose ablation of *Hdac3*. We also demonstrate that when H3atKO mice are fed HFD the FA futile cycle is less active. Our observations underline a possible mechanism operative under normal physiological conditions, which is impaired with an obesogenic diet.

In conclusion, an intrinsic crosstalk between metabolic pathways and chromatin remodeling (for example, histone acetylation) emerged by disentangling the complex phenotype of H3atKO mice and we identified HDAC3 as a molecular brake of the flux of acetyl groups from mitochondria to selective regions of chromatin in white fat. Consequently, genetic ablation of HDAC3 drives transcriptional remodeling of "hot regions" on chromatin, resulting in a different adipose phenotype, with higher FA utilization. Any cues (for example, HFD) that alter this equilibrium between chromatin remodeling and cellular metabolism may lead to a different phenotypic outcome (for example, reduced FA utilization).

Altogether, our results shed light on new facets of adipose tissue physiology, unlocking mechanisms that could be explored for new approaches in the treatment of adipose tissue dysfunction related to metabolic disorders.

## Methods

**Differentiation and infection of C3H/10T1/2 Cells**. C3H/10T1/2, Clone 8 (ATCC CCL-226) were used as a cellular model of pre-adipocytes to investigate the molecular mechanism underlying the effects observed *in vivo*. Cells were maintained in Dulbecco's modified Eagle's medium–10% FBS. To induce differentiation cells were maintained for 3 days in medium supplemented with 5 μg/ml insulin, 0.5 mM 3-isobutyl-1-methylxanthine (IBMX, Sigma Aldrich), 2 μg/ml dexamethasone (Sigma Aldrich), 5 μM rosiglitazone (Cayman Chemicals) and then switched to medium supplemented with 5 μg/ml insulin for 6 days. At the induction of differentiation cell were infected with 100 MOI scramble or shHDAC3 adenovectors (Vector BioLabs) to knockdown HDAC3.

To silence *Acly* and *Acss2*, cells were infected with 50 MOI shACLY/shACSS2 adenovectors (Vector BioLabs). To perform co-silencing of Hdac3 and Acly/Acss2,

cells at the induction of differentiation were infected with 100 MOI shHdac3 + 50 MOI shAcly or 100 MOI shHdac3 + 50 MOI shAcss2. In this case, cells infected with 150 MOI Scramble were used as negative controls.

To evaluate the role of Pparα in the metabolic rewiring consequent to Hdac3 ablation, cells were co-treated with 100 MOI Scramble or shHdac3 and vehicle (DMSO) or 10 μM GW6471 (Sigma Aldrich) at the induction of differentiation for 3 days, then medium was supplemented with 5 μg/ml insulin and vehicle/GW6471 for the following 6 days.

**$^{13}$C Tracing analysis.** At day 9 of differentiation C3H/10T1/2 cells previously infected with Scramble or shHDAC3 adenoviral vectors were treated with 100 μM [U-$^{13}$C]Palmitate (Sigma Aldrich) and 1 mM Carnitine (Sigma Aldrich) for 30 min. [U-$^{13}$C]Palmitate and Carnitine solutions were prepared as described previously[37]. Methanolic extracts of cells were analyzed by electrospray ionization flow injection analysis tandem mass spectrometry (FIA-MS/MS). For citrate, acetylCoA, palmitate, and malonylCoA each isotopomer was resolved by $m/z$ ratios and quantified by Mass Distribution Vector (MDV), as described previously[38].

Then, the percentage of MDV for each isotopomer was normalized by protein content, measured by BCA Assay (Euroclone).

**Acidic extraction of histones and mass spectrometry analysis.** At day 3 of differentiation, C3H/10T1/2 cells previously infected with Scramble or shHDAC3 adenoviral vectors were treated with 500 μM [U-$^{13}$C]Palmitate (Sigma Aldrich), 25 μM CoA (Sigma Aldrich), and 1 mM Carnitine (Sigma Aldrich) for 3 h and then pelleted. Each cellular pellet (three biological replicates for each condition) was re-suspended carefully in 7.5 ml of N-Buffer (10% Sucrose, 0.5 mM EGTA, pH 8.0, 60 mM KCl, 15 mM NaCl, 15 mM HEPES, pH 7.5, 30 μg/ml Spermidine, protease inhibitors cocktail, phosphatase inhibitors cocktail, 1 mM DTT) and 500 μl of Triton X-100 solution 8% (460 μl of N-Buffer + 40 μl Triton X-100). The samples were rolled for 10′ at 4 °C. Then each lysate was put onto a sucrose cushion (2 g of sucrose in 20 ml of N-buffer) and centrifuged at 4,000 rpm for 20′. The supernatant was removed while the nuclear pellet was washed twice with 2 ml of ice-cold PBS. Upon the addition of an equal volume of HCl 0.8 N, the nuclei were extracted overnight at 4 °C rolling. The next day each sample was centrifuged at 13,000 r.p.m. for 10′. The pellets were re-extracted in 0.4 N HCl for 4 h. All the supernatants were collected and pooled. In order to change the supernatant buffer 0.4 N HCl buffer to 0.1 M CH3COOH and finally to water, AMICON filters (MWCO 3 kDa, Millipore) were used. Protein concentration was determined using Direct Detect IR Spectrophotometer (Millipore). 40 μg of each sample were loaded on a 17.5% Sodium dodecyl sulfate polyacrylamide gel electrophoresis (SDS-PAGE). For the analysis of peptides, the bands corresponding to H3 histone were excised upon Coomassie staining, reduced with ditiothreitol, alkylated with iodoacetamide and digested with sequencing grade endoproteinase Arg-C (Roche Diagnostics). For selective analysis of acetylation of Lysine 27, the bands corresponding to H3 histone were excised, modified with acetic anhydride-d6 (SIGMA) and digested with sequencing grade trypsin (Roche Diagnostics). The peptides were extracted from gel, desalted on SCX and C/C18 home-made stage tips[39]. Peptide mixtures were subjected to nUPLC-MS/MS analysis through the Easy-nLC 1000 system coupled to Q-Exactive mass spectrometer (Thermo Fisher Scientific), using a home-made 12.5 cm column (360 μm OD, 75 μm ID, 1.9 μm Reprosil C18 AQ 120 Å resin, Dr Maisch GmbH), 0.1% formic acid in water as solvent A and 0.1% formic acid in acetonitrile as solvent B. A gradient from 0 to 45% B in 45′ was applied in order to achieve peptide separation at 300 nl/min. Full scan mass spectra were acquired on the Q-Exactive mass spectrometer in the mass range $m/z$ 300–2,000 Da with the resolution set to 70,000 at $m/z$ 400 and a target value (AGC) of $3 \times 10^6$. The 'lock-mass' option was used for accurate mass measurements. The ten most intense doubly and triply charged ions were automatically selected and fragmented by HCD with a normalized collision energy setting of 27, AGC of $1 \times 10^5$ for MS/MS. Target ions already selected for the MS/MS were dynamically excluded for 15 s. To quantify histone modifications, the area under the curve (AUC) of the extracted ion chromatograms for the modified forms relative to H3 peptides (and selectively relative to acetylated lysine 27 in H3(27–40) peptide) was calculated using a mass tolerance of 10 p.p.m. and a mass precision up to 4 decimals in the QualBrowser version 3.1.66.10. For each modified peptide, the relative abundance percentage was evaluated as a ratio between the AUC of each specific peptide over the sum of AUC of all the modified and unmodified forms of the same peptide, applying the following formula:

$$\text{Relative abundance}(\%) = \left( \frac{\text{XIC modified peptide}}{\Sigma \, \text{XIC all observed isoforms}} \right) \times 100.$$

Each modification was validated by visual inspection of the mass spectra[40].

**Quantification of lipid content.** Quantification of lipid content was performed with Oil Red O (ORO) staining: staining was eluted from wells with 1.5 ml of 100% isopropanol (10 min incubation) and absorbance was measured (500 nm, 0.5 s reading). ORO staining quantification was normalized for protein content.

**Glycerol release.** At the end of experiment, cell culture medium from C3H/10T1/2 cells grown and differentiated in 12-well plates was collected. Glycerol release in cell medium, as an index of FAs β-oxidation, was measured with Triglyceride Dosage Kit (Sentinel) and normalized for protein content.

**Gene expression.** RNA from cell cultures was isolated with TRIzol (Invitrogen) and purified using the Nucleospin RNA II kit (Macherey-Nagel). RNA from adipose tissues was isolated with Qiazol and purified using Qiazol Reagent (Qiagen), purified with commercial kit (RNeasy Lipid Tissue Mini kit, Qiagen), and quantitated with Nanodrop (Thermo Scientific, Wilmington, DE). Specific mRNA was amplified and quantitated by real time PCR, using iScriptTM One Step RT PCR for Probes (cat. # 170-8894, Bio-Rad Laboratories), following the manufacturer's instructions. Data were normalized to 36B4 mRNA and quantitated setting up a standard curve.

Samples were quantified by real-time PCR using SYBR Green or TaqMan probes on a CFX384 real-time system instrument in 384 well format (Bio-Rad Laboratories) using the iScriptTM one-step qRT-PCR kit for SYBR Green or for probes (Bio-Rad Laboratories). Primers and probes were obtained from Eurofins MWG Operon. For gene expression analysis in C3H/10T1/2 cells differentiated with Scramble or shHdac3, data from four independent experiments were included ($n = 4$). Complete list of primers is reported in Supplementary Table 3.

**RNA-seq.** RNA-seq was performed in triplicate. Each sequenced sample was a pool of 500 ng of total RNA from IngWAT of 2/3 individual mice. RNA-seq libraries were prepared with PolyA selection using 500 ng of total RNA and the Illumina TruSeq Stranded RNA reagents (Illumina; San Diego, CA, USA) on a Sciclone liquid handling robot (PerkinElmer; Waltham, MA, USA) using a PerkinElmer-developed automated script. Cluster generation was performed with the resulting libraries using the Illumina TruSeq SR Cluster Kit v3 reagents and sequenced on the Illumina HiSeq 2500 using TruSeq SBS Kit v3 reagents. Sequencing data were processed using the Illumina Pipeline Casava 1.82. Purity-filtered reads were adapter and quality trimmed with Cutadapt (v. 1.3[41], and filtered for low complexity with seq_crumbs (v. 0.1.8). Reads were aligned against Mus musculus (version GRCm38) genome using STAR (v. 2.4.2a,[42]. The number of read counts per gene locus was summarized with htseq-count (v. 0.6.1,[43]) using Mus musculus (Ensembl v. GRCm38.82) gene annotation. Statistical analysis was performed for genes in R (R version 3.2.3). Genes with low counts were filtered out according to the rule of 1 count per million (cpm) in at least 1 sample. Library sizes were scaled using TMM normalization (EdgeR package version 3.12.1[44], and log-transformed with limma voom function (Limma package version 3.26.9,[45]. Differential expression was computed with limma[46]. We used DAVID[47, 48] to highlight the biological functions enriched in differentially expressed genes. The annotated terms were grouped according to the degree of their co-association genes with the functional annotation clustering feature (Supplementary Table 5). The Group Enrichment Score, namely the geometric mean (in -log scale) of member's $p$-values in a corresponding annotation cluster, is used to rank their biological significance.

**Protein analysis.** C3H/10T1/2 cells cultured in 6-well plate and adipose tissues were lysed with SDS 2X sample buffer (20% glycerol, 4% SDS, 100 mM Tris-HCl pH 6.8, 100 mM dithiothreitol, 0.002% bromophenol blue), 1 mM protease inhibitors (Sigma), and homogenized with TissueLyser (Qiagen). Tissue lysates were separated by SDS-PAGE and transferred onto nitrocellulose membranes, and sample loading and transfer evaluated by Ponceau staining. Membranes were then blocked in 5% non fat dry milk in 1× TBS. The primary antibodies are listed in Supplementary Table 6.

HRP-conjugated goat anti-mouse (Sigma-Aldrich) and goat anti-rabbit (Cell Signaling) secondary antibodies were used for detection with chemiluminescence (ECL, Pierce).

**Chromatin immunoprecipitation.** IngWAT or C3H/10T1/2 cells infected with 100 MOI scramble or shHDAC3 adenovectors and differentiated for 9 days were cross-linked for 10 min with 1% formaldehyde. Cross-linking was stopped by addition of glycine (final concentration 125 mM). Cells were pelleted and lysed, and DNA was sheared by sonication: 8–10 pulses for 10 s for 12 h-differentiated cells and 15–20 pulses for 10 s for 9 days-differentiated cells (SLPe sonicator, Branson). H3K27ac antibody (1 μg, ab4729, Abcam) was pre-incubated with 20 μl G protein DynaBeads (Life technologies) for 1 h at room temperature, to allow the binding of the antibody to Dynabeads. Cross-linked chromatin was incubated with Dynabeads solution overnight at 4 °C. After reversing the crosslinking by incubating overnight at 65 °C, the DNA was cleaned on QIAquick gel extraction kit columns (QIAGEN). Inputs and immunoprecipitated samples were analyzed by qRT-PCR using gene-specific primers, listed in Supplementary Table 4. Data are expressed as 10% of input.

**Animal studies.** To obtain specific deletion of Hdac3 in adipose tissues, we crossed Hdac3 floxed mice in C57Bl/6J background (provided by Dr. Scott Hiebert, Vanderbilt University) with B6;FVB-Tg(Adipoq-cre) 1Evdr/J (stock # 010803, The Jackson Laboratory).

8-week-old male *Hdac3* floxed and knock out mice were fed standard diet (4RF21 GLP certificate, Mucedola), HFD (D12451, Research Diets), LFD (D12450H) for 24 weeks monitoring body weight and fasting glycemia. At the end of treatments, brown and white fat (inguinal subcutaneous and epididymal), liver and blood samples were collected from individual animals. All the experiments were performed on adult male mice (32 weeks old) in C57bl/6J background.

Triglyceride, NEFA and cholesterol levels were determined by Plasma Triglyceride Kit (Sentinel), NEFA Kit (Wako Chemicals) and by Plasma Cholesterol Kit (Sentinel). For glucose tolerance tests, mice were fasted for 16 h and glucose levels from tail vein blood were determined before and 30, 60, 90, and 120, 150 and 180 min after i.p. injection of glucose (1 g/kg) using One touch Ultra glucometer. For cold challenge test, basal rectal temperature was measured at 24 °C. Then mice were housed at 4 °C and rectal temperature was measured every 20 min. Of note, mice fed HFD were incubated at 4 °C up to 100 min because of hypothermia; on the contrary, with mice fed LFD we were able to conduct the cold challenge up to 24 h, measuring rectal temperature every hour. For collection of BAT at 4 °C, mice at LF diet were housed to 4 °C for 24 h and BAT was collected from individual animals at 4 °C.

All experiments were conducted following the regulations of the European Community (EU Directive 86/609/CEE and 2010/63/EU of the European Union) and local regulations (Italian Legislative Decree n. 116/1992 and n. 26/2014) for the care and use of laboratory animals. The Italian Ministry of Health approved the animal protocols of this study (protocols 06/2012 and 898/2015-PR).

**Histological analysis of adipose tissues**. Brown adipose tissue, subcutaneous WAT and liver were fixed with Carnoy solution (6 parts 100% Ethanol, 3 parts chloroform and 1 part glacial acetic acid) for 24 h at 4 °C. Tissues were then transferred in 100% ethanol. Tissues were embedded in paraffin and 8 μm sections were stained with hematoxylin and eosin. Images were taken at 20× magnification. Cell size of adipocytes from WATs sections were quantified using the software Photoshop CS6 (Adobe Systems Inc., San Jose, CA, USA).

**Immunohistochemistry**. Sections of WAT (8 μm) were deparaffinized, and antigen retrieval was performed with 0.05 mol/l NH$_4$Cl for 30 min, room temperature. Endogenous peroxidase activity was blocked with 1% H$_2$O$_2$ for 20 min. Blocking was performed in 1% BSA–0.1% Triton X-100 for 1 h. Anti-UCP1 (Abcam) was applied (1:300) overnight at 4 °C. Incubation with biotinylated secondary antibodies (1:3,000) followed. Histochemical reactions were performed using diaminobenzidine.

**Immunofluorescence**. Sections of subWAT (8 μm) were deparaffinized, and antigen retrieval was performed with HCl 1N for 10 min at 4 °C and HCl 2N for 10 min, room temperature and 20 min at 37 °C. Blocking was performed in 5% BSA–0.1% Triton X-100 for 1 h. Anti-ACLY (Abcam) was applied (1:200) overnight at 4 °C. Incubation with AlexaFluor 488 secondary antibodies (1:1,000) followed. Nuclear staining was performed by adding Hoechst (1:1,000) for 30 min. Nuclei containing green signal, as a measure of Acly localization in the nucleus, were counted and divided by the total number of nuclei for each section (n = 3 H3atKO mice vs. Floxed mice).

**Mass spectrometry analysis**. Inguinal and epididymal WAT from wild type and H3atKO mice were homogenized in methanol with tissue lyser (Qiagen).

For the acylcarnitines and metabolites quantification, methanolic extracts of inguinal and epididymal WAT after addition of internal standard (O-Propionyl-L-carnitine·HCl-D3, C3) were analyzed in the positive-ion mode by electrospray ionization flow injection analysis. For quantitative analysis of total FAs, methanolic extracts of liver, subWAT and viscWAT, after addition of internal standards (heneicosanoic acid, C21:0, and 13C-labeled linoleic acid, Sigma-Aldrich) were subjected to acidic hydrolysis and processed as previously described[49].

Acylcarnitine and FA quantification were performed on an API-4000 triple quadrupole mass spectrometer (AB SCIEX) coupled with a HPLC system (Agilent) and CTC PAL HTS autosampler (PAL System). The quantification of all metabolites was normalized on protein content, as reported by Cummins and collaborators[50].

**Statistical analyses**. Statistical analyses were performed using the unpaired two-tailed Student's *t*-test, one way ANOVA or two way ANOVA, with Tukeys as *post hoc* test, with GraphPad PRISM (San Diego, CA, USA).

**Data availability**. Gene expression data have been deposited in Gene Expression Omnibus (GEO) under accession code GSE90987. The authors declare that all data supporting the findings of this study are available within the article and its Supplementary Information Files or from the corresponding author upon reasonable request.

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

## Acknowledgements

We are grateful to the Genome Technologies Facility of the University of Lausanne where the RNA-seq experiment was performed. We are indebted to Dr. Greta Giordano Attianese (Centre Intégratif de Génomique, Université de Lausanne) for helpful suggestions on chromatin immunoprecipitation assays in IngWAT, and to Silvia Pedretti for the contribution in the analysis of mass spectrometry results. We thank Marco Giudici and other members of the laboratory for valuable discussion, and Elda Desiderio Pinto for administrative support. This research was supported by grants from FP7 NR-NET PITN-GA-2013-606806, PRIN 2009K7R7NA and CARIPLO Foundation 2015-0641 to M.C.

## Author contributions

A.F. conceived, planned and executed experiments, and wrote the manuscript with M.C.; R.L. executed immunofluorescence and immunohistochemistry staining on adipose tissues and performed *in vivo* studies with mice; E.F. contributed to the generation, maintenance of *Hdac3* KO mice and to *in vivo* tests; R.S. performed ChIP assays from IngWAT of mice; N.M. contributed to metabolomic analyses; G.C. executed metabolomic analyses; F.G. processed samples for RNA sequencing; B.D. supervised RNA sequencing experiments and helped manuscript corrections; A.A. and C.M. performed proteomic analysis of histones; D.C. supervised data of metabolomic analysis; E.D.F. contributed in the discussion of manuscript data; S.W.H. provided *Hdac3* floxed mice and contributed to the manuscript correction; M.C. planned the overall experimental design, supervised experiments, discussed data and wrote the manuscript with A.F.

## Additional information

**Competing interests:** The authors declare no competing financial interests.

