## [Peer Review File · Nature Communications]

Reviewers' comments:

Reviewer #1 (Expert in epigenetics and metabolism; Remarks to the Author):

In the present manuscript by Ferrari et al., entitled 'Histone deacetylase 3 is a molecular brake of white adipose tissue browning' the authors establish a link between adipocyte metabolism and epigenetics.

The manuscript is a follow-up study based on data generated in the senior author's lab analyzing the effects of deacetylase inhibitors on an obesity mouse model (Galmozzi et al., 2013). In this publication, the authors identified Hdac3 as a major player in this pathway.

In the present manuscript they follow this lead by using an adipocyte-specific knock-out of Hdac3. Overall, the present study is a very thorough analysis of the effect of Hdac3 on the metabolism of White Adipocytes (WAT) and provides some insight into a link between metabolism and epigenetics.

The differences in metabolism between floxed and Hdac3 ablation animals under different diets are very intriguing. However, it is difficult to understand why the effect on histone acetylation is very different under these conditions given that the acetyl-CoA levels in animals fed on a high-fat diet (HFD) are higher than in animals under low-fat diet (LFD). How is the futile fatty acid cycle initiated by Hdac3 ablation and LFD and is the higher histone acetylation cause or consequence? Why is acetyl-CoA shuttled towards chromatin in LFD conditions, but not under HFD conditions? The authors provide a bit of an explanation (differential expression of Vegfa) in the discussion, but generally, I find that the authors miss out on this point. Nevertheless, the manuscript describes a very interesting finding that should be of interest to the field and it should be considered for publication after addressing the comments outlined in this review.

Major specific points:

1) The combination of metabolomics and transcriptional analysis (by RT-qPCR) offers often only an indirect view on the mechanistic details of the Hdac3 ablation. The experiments outlined in Figure 3 would profit from biochemical experiments addressing the potential regulation of pyruvate dehydrogenase analyzing the phosphorylation status of the enzyme by western blotting to provide direct evidence of such regulation. In line with this, expression data for pyruvate DH kinase is given, but not for the phosphatase. Moreover, the metabolite analysis indicates very nicely that the metabolic flux is through citrate towards acetyl-CoA. To bolster this point, the authors should additionally measure pyruvate levels and exclude that under the experimental conditions pyruvate is converted to lactate.

2) To really tie in the epigenetics part of the manuscript with the metabolomics analysis the authors should perform ChIP experiments in freshly isolated tissues addressing H3K27ac levels in the enhancers of the tested genes. Ideally, the authors should attempt a whole genome approach in the conditions tested, combining ChIP-Seq for H3K27ac with RNA-Seq to make a much stronger point. Such data would provide deeper insights into the metabolic and transcriptional rewiring of the Hdac3 ablation.

3) MS-275 is a class 1 HDAC inhibitor, inhibiting at least Hdac1 with a higher efficacy than Hdac3. To be able to conclude that the effects observed in Figure 4 are due to specific inhibition of Hdac3, the authors should repeat at least the key experiment using Hdac3-specific RNAi.

Minor points:

1) The quality of western blots (particularly Fig. 1a and 2h) is very poor

2) In all western blots, H3 levels increase upon Hdac ko or Hdac inhibition. Is there any explanation for this upregulation of H3?

3) It would be nice if the graphs would be consistently headed by a description of their content -> that would make understanding of the figure much easier

4) Page 12, line 6: duplicated 'heat'

5) The authors should state the number of samples analysed by ChIP and whether the SEM stems from true biological replicates

Reviewer #2 (Expert in cellular metabolism and epigenetics; Remarks to the Author):

This manuscript by Ferrari et al examines the role of HDAC3 in WAT physiology. Deletion of HDAC3 in adipocytes was found to promote the browning of white fat. Mechanistically, authors conclude that HDAC3 deletion promotes browning by increasing the flux of acetyl groups from mitochondria to chromatin. This then increases acetylation of Pparg and Ucp1 enhancers to promote browning. While the model proposed is appealing, most of the core conclusions are not adequately supported by the data presented. Substantial additional work would be needed to test whether the proposed mechanism is correct.

Major points:

1. Regarding the total histone acetylation blots, I would urge authors to re-do the histone blots use acid-extracted histones. The loading of total H3 is very uneven in these samples (Fig 1 and 4) and it is easy to draw incorrect conclusions if histones have been unevenly extracted from samples. For example, the complete deficiency of H3K14ac in WT mice or all acetylation in DMSO-treated cells makes me question the sample quality.

2. While browning-related gene expression is shown to be higher at baseline in WAT in the H3atKO, what happens during cold exposure? Additionally, since HDAC3 is deleted in both brown and white fat, how do authors know that effects on cold challenge are due to white fat browning? Gene expression in BAT is shown at baseline, but it would be helpful to see histology and gene expression during cold exposure as well.

3. Authors profile gene expression and metabolites in the adipose tissue of WT vs H3atKO mice. They make conclusions about metabolic flux, but this is unwarranted, since the data is all steady state.

4. Authors use cell lines to gain more mechanistic insight into the phenotype of the mouse model. However, treating with an HDAC inhibitor is not the same as deleting HDAC3. Authors should repeat the cell line experiments with a genetic HDAC3 deletion/ knockdown.

5. Most importantly, a central conclusion of the paper is that the flux of acetyl groups from mitochondria to chromatin underlie the phenotype. Yet, nowhere in the paper do authors actually test this. It is essential that authors conduct isotope labeling experiments in a cell line with genetic HDAC3 manipulation to test their model. Alternatively (and perhaps even better), it may be possible to do this using primary adipocytes from the H3atKO mice. For example, authors could use ¹³C- or ¹⁴C-labeled palmitate and test 1) whether increased fatty acid oxidation is observed and 2) whether carbon from fatty acids actually ends up on histone acetyl groups. In case authors don't have access to a proteomics facility to assess histone acetylation by MS, they can label cells with ¹⁴C-palmitate, extract histones, use scintillation counting.

6. It is essential to test whether the increase in Pparg and Ucp1 enhancer acetylation is due to increased acetyl-CoA coming from mitochondria. Since this would be dependent on ACLY, authors could use knockdown or inhibition of ACLY to determine if that suppresses the increase in enhancer acetylation.

7. Authors claim that there is a futile cycle of fatty acid synthesis and oxidation. Again, authors should test whether both fatty acid synthesis and oxidation are both elevated using isotope tracer experiments.

Minor points:

1. Are the effects of HDAC3 deletion on histone modification specific to H3K14ac and H3K27ac? Why did authors only look at these marks? If the proposed mechanism is correct, I would expect this to be a fairly general phenomenon regulating multiple histone acetyl marks.
2. Authors should make it more clear why they don't think the hyperacetylation is due to direct effects of loss of HDACs but instead through an indirect mechanism involving acetyl-CoA production?
3. Fig 3: which Pk isoform was tested?

Reviewer #3 (Expert in adipose tissue biology; Remarks to the Author):

A. Ferrari and colleagues study the role of HDAC3 in the white adipose tissue browning through epigenetic regulation of thermogenic genes. They also proposed a model of HDAC3 inactivation-triggered futile cycle of lipid metabolism boosting browning of white adipose tissue. The study is a follow-up of the authors' previous work which reveals the regulatory role of class I HDACs and histone acetylation in browning. But in the new study, the lack of observable phenotype caused by HDAC3 inactivation in the diet-induced obese model apparently weakened the physiological significance of HDAC3 in browning and lowered its potential as a druggable target, especially in the context that plenty of evidence has proved that other HDACs (HDAC1, HDAC2, HDAC9, Sirt1, etc.) mediate and/or contribute to browning even in obese animal models. Besides, one of the major conclusion made by the authors is that HDAC3 mediate the histone deacetylation upon the enhancer region of key thermogenic genes such as Ppar γ and Ucp1. HDAC-mediated histone deacetylation is the key mechanism of silencing gene transcription by compacting chromatin and making chromatin less accessible to transcriptional activators. As described by the authors in the manuscript, HDAC3 inactivation leads to a global increase of acetylation of histone within the genome. So presumably, all suppressed gene locus in the white adipocytes would be "opened" and accessible to transcriptional activation. Thermogenic genes are clearly the most silenced or suppressed genes in white adipocytes, thus there is no surprise that they're robustly upregulated. It'd be advisable that the authors provide additional data about (1) A detailed description of brown adipose tissue in the HDAC3 knockout animal, especially the suppressed "white-selective", pro-storage genes. This could be helpful and informative to determine whether the HDAC3-inactivation is selective to regulate thermogenic genes or only a general mechanism of releasing the suppression of silenced genes. (2) To determine the selectiveness of epigenetic regulation of thermogenic genes by HDAC3 in the white adipose tissue. Addressing this point would be highly informative in determining the specificity and importance of HDAC3 in regulating browning and the significance of the study.

Apart from the suggestions described above, other more detailed questions are listed as follows:

1. There is no clearly evidence linking the HDAC3-inactivation induced browning to the naturally physiological regulation of browning like cold exposure or β 3 adrenergic agonism. Is histone acetylation status altered upon cold exposure or β 3 adrenergic agonist administration? Is the alteration mediated by HDAC3? Is HDAC3 released from the enhancer regions of Ppar γ and Ucp1 upon cold exposure or β 3 adrenergic agonist administration?
2. In Figure 4, the authors used the class I HDAC inhibitor to "phenocopy" HDAC3 inactivation. But as reported and described by the authors, HDAC inhibitor is inducing browning and HDAC1/2 are at least partially responsible. So the data in Figure 4 is barely relevant to this study or supportive to determine the function of HDAC3 in browning. Instead, it'd be more informative and relevant if the authors use primary cell isolated from the HDAC3 knockouts to reproduce and confirm the findings in vitro.
3. As shown in the manuscript, a large number of adipocyte genes are upregulated after HDAC3 inactivation, including general adipose genes, oxidative genes, and browning markers. Most of the regulated genes, if not all, are targets of Ppar γ , including Ppar γ itself. In another study by Dr. D. Accili's lab has indicated that HDAC (Sirt1) modulates the activity of Ppar γ via post-translational

acetylation regulation of the protein. Thus, it'd be interesting to test the possibility that HDAC3 regulating Pparg activity above transcription. This could broaden the mechanistic insight of the study.

4. Another important and interesting point proposed by authors here is the enhanced β -oxidation and futile cycle of lipid metabolism which provide the acetyl-CoA as the substrate of histone acetylation robustly occurring after HDAC3 inactivation. And the authors ascribed the absence of notable phenotype of HDAC3 knockouts in the diet-induced obesity model to the impairment of lipolysis/ β -oxidation caused by high fat feeding. Yet, (1) There is no direct measurement of β -oxidation rate. (2) There is no direct evidence supporting the requirement of the futile cycle as the major source of acetylation substrate. Knockdown/knockout ACLY in the HDAC3-knockout cells could be one of the experiments to determine the requirement.

5. Please include Ucp1 expression level in the gonadal data in Supplemental Figure 2b.

6. Data in Figure 4 indicate inhibition of class I HDACs during early differentiation stage could cause adipocyte browning. It'd be interesting to know if inhibition of class I HDACs or special inactivation of HDAC3 in differentiated mature adipocytes could cause browning as well? This is potentially a clue of whether HDAC inhibition-mediated browning is more relevant to beige adipogenesis or thermogenic activation of "dormant" white adipocytes under physiological conditions.

Reviewer #4 (Expert in metabolomics; Remarks to the Author):

The manuscript by Ferrari et al. evaluates the role of histone deacetylase 3 (HDAC3) on the function of white adipose tissue (WAT) and energy metabolism. The authors found that a biological function of HDAC3 in WAT is to mediate WAT browning. The deletion of Hdac3 in adipose tissue altered the metabolic signature of WAT and was concluded to occur by increasing the flux of carbon from the mitochondria to chromatin, which results in increased acetylation of enhancers in Pparg and Ucp1, key components mediating browning and thermogenesis. The authors have investigated an important problem in the epigenetic regulation of adipose function and energy metabolism, and have reached novel conclusions which might provide novel therapeutic targets for metabolic disease. However, the impact of HDAC3 on adipose physiology is modest. Moreover, certain conclusions could be better supported with better evidence.

Major comments:

1. In the in vivo model with deletion of Hdac3 in adipose tissue, Hdac3 was clearly reduced in BAT (supplementary Fig.1). With the effect on browning of WAT, one might expect an effect on BAT-mediated thermogenesis, since the KO mice shows better endurance upon cold exposure. Although the authors have examined the mRNA levels of gene expression in BAT, the impact on function of BAT could be further explored to confirm that indeed BAT is not accounting for the increased thermogenesis under the cold conditions, such as by biochemical quantification of lipids in the tissue, adipocyte size and et al.

2. As the lack of impact on WAT in HFD-fed mice, one might question whether HFD itself affects browning in WAT through HDAC3?

3. The authors claimed that there is increased lipogenesis and lipolysis in WAT, which composed a futile cycle in both animal models and in vitro cell models, mainly based on mRNA data. To support these claims, additional functional analysis are required such as the examination the lipogenesis in WAT and cells by radioactive labeling. The increased lipolysis could be better served by monitor the kinetics of the release of glycerol and fatty acids in culture media (for cells or tissue explants). The levels of glycerol and fatty acids in plasma would likely provide some insights.

Minor

1. The size of the adipocytes could be quantified in a more meaningful way by (μm^2) instead of pixels

2. Fig 1a - The western blot needs to be improved to the level of publication quality. Equal loading across the lanes is essential.

NCOMMS-16-01290 "Histone deacetylase 3 is a molecular brake of white adipose tissue browning" – Point-by-point Response to Reviewers

On behalf of all authors of the manuscript n. NCOMMS-16-01290 "**Histone deacetylase 3 is a molecular brake of white adipose tissue browning**", I would like to thank all reviewers for providing us with very constructive comments to improve the story.

Following the comments of the reviewers we performed a lot of new experiments that helped us respond all the comments raised by the four reviewers. The execution of these experiments required a lot of work and took several months to complete them. According to the new results we have partially refocused the manuscript and the interpretation of the mechanisms underlying our observations in this interesting knock out model. For this reason, we also modified the title to "**Histone deacetylase 3 is a molecular brake of the metabolic rewiring that sustains browning of white adipose tissue**", to emphasize the important concept of metabolic rewiring linked to browning of white fat.

I am now submitting a revised version of the manuscript that has been vastly improved following the useful comments and suggestions of the reviewers.

Following is the point-by-point response to concerns raised by each reviewer.

Reviewer #1

General Comment:

In the present manuscript by Ferrari et al., entitled 'Histone deacetylase 3 is a molecular brake of white adipose tissue browning' the authors establish a link between adipocyte metabolism and epigenetics.

The manuscript is a follow-up study based on data generated in the senior author's lab analyzing the effects of deacetylase inhibitors on an obesity mouse model (Galmozzi et al., 2013). In this publication, the authors identified Hdac3 as a major player in this pathway.

In the present manuscript they follow this lead by using an adipocyte-specific knock-out of Hdac3. Overall, the present study is a very thorough analysis of the effect of Hdac3 on the metabolism of White Adipocytes (WAT) and provides some insight into a link between metabolism and epigenetics.

The differences in metabolism between floxed and Hdac3 ablation animals under different diets are very intriguing. However, it is difficult to understand why the effect on histone acetylation is very different under these conditions given that the acetyl-CoA levels in animals fed on a high-fat diet (HFD) are higher than in animals under low-fat diet (LFD). How is the futile fatty acid cycle initiated by Hdac3 ablation and LFD and is the higher histone acetylation cause or consequence? Why is acetyl-CoA shuttled towards chromatin in LFD conditions, but not under HFD conditions? The authors provide a bit of an explanation (differential expression of Vegfa) in the discussion, but generally, I find that the authors miss out on this point. Nevertheless, the manuscript describes a very interesting finding that should be of interest to the field and it should be considered for publication after addressing the comments outlined in this review.

It is difficult to understand why the effect on histone acetylation is very different under these conditions given that the acetyl-CoA levels in animals fed on a high-fat diet (HFD) are higher than in animals under low-fat diet (LFD). How is the futile fatty acid cycle initiated by Hdac3 ablation and LFD and is the higher histone acetylation cause or consequence? Why is acetyl-CoA shuttled towards chromatin in LFD conditions, but not under HFD conditions? The authors provide a bit of an explanation (differential expression of Vegfa) in the discussion, but generally, I find that the authors miss out on this point.

Response: We demonstrated that the establishment of Hdac3 KO phenotype is related to a metabolic reprogramming that involves several pathways (glucose and lipid metabolism). This metabolic rewiring is strictly linked to events occurring at selective chromatin regions (e.g. *Ppara*, *Pparg*, *Ucp1*), highlighting that histone acetylation is an integral part of metabolism linked to transcriptional regulation. In this scenario, high fat diet feeding perturbs the equilibrium of this metabolic flux observed in Hdac3 KO mice fed low fat diet, leading to a different phenotypic outcome. Some of the key metabolic events induced by Hdac3 ablation are in fact blunted by exposure to dietary lipid overload (see figures 7 and 8 and discussion).

Major specific points:

1) The combination of metabolomics and transcriptional analysis (by RT-qPCR) offers often only an indirect view on the mechanistic details of the Hdac3 ablation. The experiments outlined in Figure 3 would profit from biochemical experiments addressing the potential regulation of pyruvate dehydrogenase analyzing the phosphorylation status of the enzyme by western blotting to provide direct evidence of such regulation. In line with this, expression data for pyruvate DH kinase is given, but not for the phosphatase. Moreover, the metabolite analysis indicates very nicely that the metabolic flux is through citrate towards acetyl-CoA. To bolster this point, the authors should additionally measure pyruvate levels and exclude that under the experimental conditions pyruvate is converted to lactate.

Response: We thank the reviewer for this important comment. We have performed western blot of PDH phosphorylation status and we found strongly increased in phosphorylation on Ser293 and Ser300 of PDH in adipose tissue from mice lacking Hdac3 fed low fat diet, consistent to elevated expression of *Pdk4* gene (Figure 3f,g in the revised manuscript). Interestingly, these modifications were abrogated with mice fed HFD, giving important insights on how a different nutritional regimen could interfere with the Hdac3KO-mediated phenotype (see figure 8 and discussion).

We also measured pyruvate and lactate levels in WAT from floxed and Hdac3KO mice fed low fat diet, and we found reduced levels of these metabolites in KO mice, indicating that pyruvate is mainly shuttled to the anaplerotic reaction (pyruvate carboxylase) producing oxaloacetate. These data are consistent with increased expression of *Pcx*, the gene encoding pyruvate carboxylase, with increased expression of *Pdk4* (Figure 3f in the revised manuscript) and with enhanced phosphorylation of PDH (Figure 3g in the revised manuscript).

2) To really tie in the epigenetics part of the manuscript with the metabolomics analysis the authors should perform ChIP experiments in freshly isolated tissues addressing H3K27ac levels in the enhancers of the tested genes. Ideally, the authors should attempt a whole genome approach in the conditions tested, combining ChIP-Seq for H3K27ac with RNA-Seq to make a much stronger point. Such data would provide deeper insights into the metabolic and transcriptional rewiring of the Hdac3 ablation.

Response: As suggested by reviewer we performed RNA-seq experiments in tissues from floxed mice and KO mice on low fat diet (Figure 2a, b in the revised manuscript). Our data obtained with RNA-seq are absolutely in line with the results obtained with the multiple approaches throughout the manuscript (i.e., metabolic rewiring and browning of inguinal WAT in Hdac3 KO mice vs. floxed mice fed low fat diet). As for the request to perform ChIP-seq, although we do recognize that such experiment would be highly valuable, on the other hand there are technical hurdles that should be considered. To perform ChIP-seq in WAT, high amount of fat would be necessary, due to the particularly low yield of chromatin from this tissue. We discussed this experiment with our collaborators at the Center for Integrative Genomics University of Lausanne, who are expert in this technique and have recently set up protocols for ChIP-seq in adipose tissue. They estimated that 60 mice from each experimental group would be required. Considering four experimental groups at least in duplicate, we would need around 120 floxed mice and 120 Hdac3 KO mice. This poses serious problems to obtain the ethical permit from the Italian authorities, due to the tight regulations entered in force recently (see the 3 Rs for animal use in scientific experiments, EU directive 63/2010 and national law D.Lgs. 26/2014). Furthermore, the time required to obtain such high number of mice would be very long, making this experiment extremely difficult to plan in a reasonable time frame.

In order to get useful information about epigenetic and transcriptional regulation mediated by Hdac3 we performed ChIP PCR analysis in C3H10T1/2 adipocytes silenced with HDAC3 shRNA and in adipose tissues from floxed and Hdac3 KO mice. This approach allowed us to obtain further insights into the mechanisms underlying the phenotype of Hdac3 KO mice (Figure 5 in the revised manuscript).

3) MS-275 is a class 1 HDAC inhibitor, inhibiting at least Hdac1 with a higher efficacy than Hdac3. To be able to conclude that the effects observed in Figure 4 are due to specific inhibition of Hdac3, the authors should repeat at least the key experiment using Hdac3-specific RNAi.

Response: All results in cell cultures included in the new version of the manuscript have been obtained in C3H10T1/2 adipocytes silenced with scramble/HDAC3 shRNAs. Consequently, we decided to remove previous data obtained with MS-275.

Minor points:

1) The quality of western blots (particularly Fig. 1a and 2h) is very poor

Response: The indicated western blots were repeated and now the quality is much better.

2) In all western blots, H3 levels increase upon Hdac ko or Hdac inhibition. Is there any explanation for this upregulation of H3?

Response: These western blots were repeated with acidic extractions of histones leading to different conclusions as opposed to the original submission.

3) It would be nice if the graphs would be consistently headed by a description of their content -> that would make understanding of the figure much easier

Response: We modified graphs including headings as suggested.

4) Page 12, line 6: duplicated 'heat'

Response: Amended.

5) The authors should state the number of samples analysed by ChIP and whether the SEM stems from true biological replicates

Response: Samples analyzed by ChIP were true biological replicates and the “n” was specified in figure legends.

Reviewer #2

General comment:

This manuscript by Ferrari et al examines the role of HDAC3 in WAT physiology. Deletion of HDAC3 in adipocytes was found to promote the browning of white fat. Mechanistically, authors conclude that HDAC3 deletion promotes browning by increasing the flux of acetyl groups from mitochondria to chromatin. This then increases acetylation of Pparg and Ucp1 enhancers to promote browning. While the model proposed is appealing, most of the core conclusions are not adequately supported by the data presented. Substantial additional work would be needed to test whether the proposed mechanism is correct.

Major points:

1. Regarding the total histone acetylation blots, I would urge authors to re-do the histone blots use acid-extracted histones. The loading of total H3 is very uneven in these samples (Fig 1 and 4) and it is easy to draw incorrect conclusions if histones have been unevenly extracted from samples. For example, the complete deficiency of H3K14ac in WT mice or all acetylation in DMSO-treated cells makes me question the sample quality.

Response: We are grateful to this reviewer for pointing out this important issue. We performed the acidic extraction of histone proteins from adipose tissue of floxed and KO mice and from C3H10T1/2 adipocytes silenced with scramble/HDAC3 shRNAs. We performed western blots with Ab against pan acetyl H3 and H3K27ac and we detected no

differences in global acetylation profile in mice and cells lacking HDAC3 (Supplementary Figure 10 in the revised manuscript).

2. While browning-related gene expression is shown to be higher at baseline in WAT in the H3atKO, what happens during cold exposure? Additionally, since HDAC3 is deleted in both brown and white fat, how do authors know that effects on cold challenge are due to white fat browning? Gene expression in BAT is shown at baseline, but it would be helpful to see histology and gene expression during cold exposure as well.

Response: To address this point we incubated floxed and Hdac3KO mice in the cold and then collected tissues for gene expression analysis. We found that Hdac3 ablation recapitulates, at least in part, the transcriptional program elicited by cold exposure in IngWAT (Supplementary Figure 6 in the revised manuscript). In addition, with the tissues obtained from these mice kept in the cold we found that the thermogenic program is not enhanced in BAT in response to cold in KO mice vs. floxed mice (Supplementary Figure 3h in the revised manuscript), suggesting that the major changes in response to cold challenge could be ascribed mainly to WAT browning in Hdac3 K mice.

3. Authors profile gene expression and metabolites in the adipose tissue of WT vs H3atKO mice. They make conclusions about metabolic flux, but this is unwarranted, since the data is all steady state.

Response: To provide evidence of the flux of acetyl groups, we have performed isotope labeling experiments with ¹³C-palmitate to confirm fatty acid β-oxidation occurring in response to Hdac3 shRNA (Figure 4l-n in the revised manuscript).

4. Authors use cell lines to gain more mechanistic insight into the phenotype of the mouse model. However, treating with an HDAC inhibitor is not the same as deleting HDAC3. Authors should repeat the cell line experiments with a genetic HDAC3 deletion/knockdown.

Response: All results in cell cultures included in the new version of the manuscript have been obtained in C3H10T1/2 adipocytes silenced with scramble/HDAC3 shRNAs (see Figure 4 in the revised manuscript for protocol details).

5. Most importantly, a central conclusion of the paper is that the flux of acetyl groups from mitochondria to chromatin underlie the phenotype. Yet, nowhere in the paper do authors actually test this. It is essential that authors conduct isotope labeling experiments in a cell line with genetic HDAC3 manipulation to test their model. Alternatively (and perhaps even better), it may be possible to do this using primary adipocytes from the H3atKO mice. For example, authors could use 13C- or 14C-labeled palmitate and test 1) whether increased fatty acid oxidation is observed and 2) whether carbon from fatty acids actually ends up on histone acetyl groups. In case authors don't have access to a proteomics facility to assess histone acetylation by MS, they can label cells with 14C-palmitate, extract histones, use scintillation counting.

Response: We have verified and demonstrated 1) increased fatty acid β -oxidation (Figure 4l-n in the revised manuscript) and 2) the incorporation of ^{13}C -acetate derived from ^{13}C -palmitate into histone proteins in C3H/10T1/2 through proteomic analysis (Supplementary Figure 10c, d in the revised manuscript).

6. It is essential to test whether the increase in *Pparg* and *Ucp1* enhancer acetylation is due to increased acetyl-CoA coming from mitochondria. Since this would be dependent on ACLY, authors could use knockdown or inhibition of ACLY to determine if that suppresses the increase in enhancer acetylation.

Response: To answer this point we knocked down *Acly* in C3H10T1/2 adipocytes. Consistent to results published by Wellen et al. (Science 2009, doi: 10.1126/science.1164097), we found that ACLY silencing blocked adipogenesis (Figure 6c in the revised manuscript). Most importantly, co-silencing of ACLY and HDAC3 completely abrogated the effects of HDAC3 inactivation, since expression analysis showed that genes that were strongly induced upon HDAC3 silencing (*Ucp1*, *Ppara*, *Pparg*, *Glut4*, *Pdk4*, *Pcx*, *Atgl*, *Acadl*, *Fasn*), were significantly reduced in cells lacking ACLY as well as in cells lacking both HDAC3 and ACLY (Figure 6d in the revised manuscript). ChIP analysis showed significant reduction of H3K27 acetylation at -5 kb of *Ucp1* enhancer and at *Ppara* putative regulatory regions in co-silenced cells compared to shHDAC3 treated cells (Figure 6e in the revised manuscript). Likewise, 50% reduction of H3K27 acetylation was observed at *Pparg* enhancer (Figure 6e in the revised manuscript).

7. Authors claim that there is a futile cycle of fatty acid synthesis and oxidation. Again, authors should test whether both fatty acid synthesis and oxidation are both elevated using isotope tracer experiments.

Response: In cells incubated with ^{13}C -palmitate (uniformly labeled), we measured the formation of ^{13}C (+2) palmitate that can be originated only from the incorporation of ^{13}C (+2) acetyl-CoA. We found it increased in adipocytes silenced with HDAC3 shRNA (Figure 4n in the revised manuscript). We also found that ^{13}C (+1) acetyl-CoA was higher in cells silenced with HDAC3 shRNA (Figure 4l in the revised manuscript). Altogether, these results with isotope tracer confirm that both fatty acid synthesis and β -oxidation are operative concomitantly in HDCA3 knock down adipocytes.

Minor points:

1. Are the effects of HDAC3 deletion on histone modification specific to H3K14ac and H3K27ac? Why did authors only look at these marks? If the proposed mechanism is correct, I would expect this to be a fairly general phenomenon regulating multiple histone acetyl marks.

Response: To get a wide view of global H3 acetylation, we performed western blot analyses with pan-acetyl H3 antibody besides that with H3K27ac antibody on acid histone extracts from adipose tissues of floxed and KO mice and from C3H10T1/2 adipocytes silenced with scramble/HDAC3 shRNAs. The results of these new western blots indicate no difference in

the acetylation levels of HDAC3 KO/KD vs. normal tissues/cells (Supplementary Figure 10 in the revised manuscript).

2. Authors should make it more clear why they don't think the hyperacetylation is due to direct effects of loss of HDACs but instead through an indirect mechanism involving acetyl-CoA production?

Response: We apologize for not explaining well. We believe that hyperacetylation of some specific chromatin regions (i.e., *Pparg*, *Ppara* and *Ucp1*) is a direct consequence of loss of HDAC3. The enhanced expression of these three genes is critical to regulate the entire downstream program. Notably, histone acetylation is linked to acetyl-CoA from fatty acid β -oxidation through citrate shuttling from mitochondria to the nucleus where ACLY catalyzes its cleavage to oxaloacetate and acetyl-CoA required for histone acetylation.

3. Fig 3: which Pk isoform was tested?

Response: The isoform is Pkm, which is expressed at fairly high levels in both BAT and WAT (see also <http://biogps.org/#goto=genereport&id=18746>)

Reviewer #3

General comment:

A. Ferrari and colleagues study the role of HDAC3 in the white adipose tissue browning through epigenetic regulation of thermogenic genes. They also proposed a model of HDAC3 inactivation-triggered futile cycle of lipid metabolism boosting browning of white adipose tissue. The study is a follow-up of the authors' previous work which reveals the regulatory role of class I HDACs and histone acetylation in browning. But in the new study, the lack of observable phenotype caused by HDAC3 inactivation in the diet-induced obese model apparently weakened the physiological significance of HDAC3 in browning and lowered its potential as a druggable target, especially in the context that plenty of evidence has proved that other HDACs (HDAC1, HDAC2, HDAC9, Sirt1, etc.) mediate and/or contribute to browning even in obese animal models.

Response: We thank this reviewer for bringing up this important issue. Regarding this point, we would like to emphasize that we did not propose HDAC3 as a druggable target, rather we wanted to provide novel knowledge to deepen our understanding of adipose tissue physiology. Eventually, this will help explore new therapeutic avenues related to the mechanistic role of HDAC3 in adipose tissue metabolism, without implying that HDAC3 *per se* could be a drug target. Although there are reports showing that HDAC3 inhibition improves glycemia and insulin secretion in obese diabetic rats (Diabetes Obes Metab, 2015 vol. 17(7):703-707), it is still too premature to establish HDAC3 as a drug target in diabetes/obesity.

Furthermore, the new results with mice exposed to cold included in the revised version of the manuscript seem to suggest a possible role of HDAC3 in the response to cold (Supplementary Figure 6 in the revised manuscript). In fact, it is noteworthy that the loss of

browning in inguinal fat of HDAC3 KO mice, made obese with high fed diet, seems to recapitulate the impaired thermogenesis typically observed in overweight/obese individuals as well as in rodent models of obesity. Altogether, these considerations suggest a possible role of HDAC3 in the adaptive response of adipose tissue to changes in external temperature, revealing a novel function of HDAC3 in the physiology of adipose tissue. Thus, we believe that these observations provide a strong indication of the physiological significance of HDAC3 in WAT browning and metabolic rewiring.

Besides, one of the major conclusion made by the authors is that HDAC3 mediate the histone deacetylation upon the enhancer region of key thermogenic genes such as Ppar γ and Ucp1. HDAC-mediated histone deacetylation is the key mechanism of silencing gene transcription by compacting chromatin and making chromatin less accessible to transcriptional activators. As described by the authors in the manuscript, HDAC3 inactivation leads to a global increase of acetylation of histone within the genome. So presumably, all suppressed gene locus in the white adipocytes would be "opened" and accessible to transcriptional activation. Thermogenic genes are clearly the most silenced or suppressed genes in white adipocytes, thus there is no surprise that they're robustly upregulated. It'd be advisable that the authors provide additional data about

(1) A detailed description of brown adipose tissue in the HDAC3 knockout animal, especially the suppressed "white-selective", pro-storage genes. This could be helpful and informative to determine whether the HDAC3-inactivation is selective to regulate thermogenic genes or only a general mechanism of releasing the suppression of silenced genes.

Response: To address this point we performed transcriptome analysis in inguinal WAT with RNA-seq. As expected, we found that some genes belonging to functional annotations related to mitochondria, fatty acid metabolism and Krebs cycle were upregulated in Hdac3 KO mice. On the other hand, the expression of other genes was reduced in inguinal WAT upon Hdac3 ablation (Figure 2a in the revised manuscript). These results prove that ablation of this transcriptional repressor did not de-repress globally all transcriptome, rather it resulted in **selective** up- and down-regulation of different subsets of genes. We believe that the common view whereby ablation/inhibition of HDACs leads to general and unspecific upregulation of gene transcription is misleading.

Furthermore, we analyzed the expression of "white selective/pro storage" genes (Retn, Ednra, Serpina3k, Psat1) in BAT to assess whether Hdac3 ablation unselectively de-repressed expression of genes typically not expressed in brown adipose tissue. None of these markers was increased in H3atKO mice (Supplementary Figure 3e in the revised manuscript).

(2) To determine the selectiveness of epigenetic regulation of thermogenic genes by HDAC3 in the white adipose tissue. Addressing this point would be highly informative in determining the specificity and importance of HDAC3 in regulating browning and the significance of the study.

Response: See previous point. Moreover, to further address this point, we incubated floxed and Hdac3KO mice in the cold and then collected tissues for gene expression analysis. We found that Hdac3 ablation recapitulates, at least in part, the transcriptional program elicited

by cold exposure in IngWAT (Supplementary figure 6 in the revised manuscript). In addition, with the tissues obtained from these mice kept in the cold we found that the thermogenic program is not enhanced in BAT in response to cold in KO mice vs. floxed mice (Supplementary figure 6 in the revised manuscript).

Apart from the suggestions described above, other more detailed questions are listed as follows:

1. There is no clearly evidence linking the HDAC3-inactivation induced browning to the naturally physiological regulation of browning like cold exposure or $\beta 3$ adrenergic agonism. Is histone acetylation status altered upon cold exposure or $\beta 3$ adrenergic agonist administration? Is the alteration mediated by HDAC3? Is HDAC3 released from the enhancer regions of *Ppar γ* and *Ucp1* upon cold exposure or $\beta 3$ adrenergic agonist administration?

Response: We agree with this reviewer that ideally it would be great to verify whether HDAC3 is released from the enhancer regions of *Ppar γ* and *Ucp1* upon cold exposure or $\beta 3$ adrenergic agonist administration. However, CHIP assay of HDAC3 in adipose tissue or in adipocyte cell lines is extremely difficult and is technically very challenging. We have been trying to set up and optimize the protocol with different Ab vs. HDAC3 but unfortunately so far we have been unsuccessful. Furthermore, to perform CHIP-seq in WAT, high amount of fat would be necessary, due to the particularly low yield of chromatin from this tissue. We discussed this experiment with our collaborators at the Center for Integrative Genomics University of Lausanne, who are expert in this technique and have recently set up protocols for CHIP-seq in adipose tissue. They estimated that 60 mice from each experimental group would be required. Considering four experimental groups at least in duplicate, we would need around 120 floxed mice and 120 *Hdac3* KO mice. This poses serious problems to obtain the ethical permit from the Italian authorities, due to the tight regulations entered in force recently (see the 3 Rs for animal use in scientific experiments, EU directive 63/2010 and national law D.Lgs. 26/2014). Furthermore, the time required to obtain such high number of mice would be very long, making this experiment extremely difficult to plan in a reasonable time frame.

To investigate the link between HDAC3 inactivation-induced browning and the physiological regulation of browning, we incubated floxed and *Hdac3*KO mice in the cold and then collected tissues for gene expression analysis. We found that *Hdac3* ablation recapitulates, at least in part, the transcriptional program elicited by cold exposure in IngWAT.

2. In Figure 4, the authors used the class I HDAC inhibitor to "phenocopy" HDAC3 inactivation. But as reported and described by the authors, HDAC inhibitor is inducing browning and HDAC1/2 are at least partially responsible. So the data in Figure 4 is barely relevant to this study or supportive to determine the function of HDAC3 in browning. Instead, it'd be more informative and relevant if the authors use primary cell isolated from the HDAC3 knockouts to reproduce and confirm the findings in vitro.

Response: We take the point raised by this reviewer. However, to comply with the tight regulations entered in force recently in Italy and reduce the number of mice used for experimentation (see the 3 Rs for animal use in scientific experiments), we performed the

cell culture experiments suggested by this reviewer in C3H10T1/2 adipocytes and silenced them with scramble/HDAC3 shRNAs.

3. As shown in the manuscript, a large number of adipocyte genes are upregulated after HDAC3 inactivation, including general adipose genes, oxidative genes, and browning markers. Most of the regulated genes, if not all, are targets of Ppar γ , including Ppar γ itself. In another study by Dr. D. Accili's lab has indicated that HDAC (Sirt1) modulates the activity of Ppar γ via post-translational acetylation regulation of the protein. Thus, it'd be interesting to test the possibility that HDAC3 regulating Ppar γ activity above transcription. This could broaden the mechanistic insight of the study.

Response: At this regard, it has been reported that treatment of 3T3-L1 adipocytes with an HDAC3 inhibitor increases acetylation status of PPAR γ . We do not exclude the possibility of a similar effect in our knock out (Jiang, X., Ye, X., Guo, W., Lu, H., and Gao, Z. (2014). Inhibition of HDAC3 promotes ligand-independent PPAR γ activation by protein acetylation. *Journal of Molecular Endocrinology* 53, 191-200). However, although we recognize the potential, the study of the possible acetylation status of factors other than histones would be beyond the scope of this manuscript and would make the story too complicated. Therefore, at this stage we rather stay focused on histone acetylation status in our KO model.

4. Another important and interesting point proposed by authors here is the enhanced β -oxidation and futile cycle of lipid metabolism which provide the acetyl-CoA as the substrate of histone acetylation robustly occurring after HDAC3 inactivation. And the authors ascribed the absence of notable phenotype of HDAC3 knockouts in the diet-induced obesity model to the impairment of lipolysis/ β -oxidation caused by high fat feeding. Yet, (1) There is no direct measurement of β -oxidation rate. (2) There is no direct evidence supporting the requirement of the futile cycle as the major source of acetylation substrate. Knockdown/knockout ACLY in the HDAC3-none cells could be one of the experiments to determine the requirement.

Response: We acknowledge this comment made by this reviewer. To provide evidence of the flux of acetyl groups, we have performed isotope labeling experiments with ^{13}C -palmitate to confirm fatty acid β -oxidation occurring in response to Hdac3 shRNA (Figure 4l,m in the revised manuscript).

Moreover, in cells incubated with ^{13}C -palmitate (uniformly labeled), we also measured the formation of ^{13}C (+2) palmitate that can be originated only from the incorporation of ^{13}C (+2) acetyl-CoA, as a read out of *de novo* fatty acid synthesis (Figure 4n in the revised manuscript). We have also verified the incorporation of ^{13}C -acetate into histone proteins in C3H/10T1/2 through proteomic analysis (Supplementary Figure 10c,d in the revised manuscript).

We have also knocked down Acl γ in C3H10T1/2 adipocytes. Consistent with results published by Wellen et al. (*Science* 2009, doi: 10.1126/science.1164097), we found that ACLY silencing blocked adipogenesis. Most importantly, co-silencing of ACLY and HDAC3 completely abrogated the effects of HDAC3 inactivation, since expression analysis showed that genes that were strongly induced upon HDAC3 silencing (*Ucp1*, *Ppara*, *Pparg*, *Glut4*, *Pdk4*, *Pcx*, *Atgl*, *Acadl*, *Fasn*), were significantly reduced in cells lacking ACLY as well as in cells lacking both HDAC3 and ACLY (Figure 6d in the revised manuscript). Finally, ChIP

analysis showed significant reduction of H3K27 acetylation at the -5 kb *Ucp1* enhancer and at *Ppara* putative regions in co-silenced cells compared to shHDAC3 treated cells, and 50% reduction of H3K27ac was observed at *Pparg* enhancer (Figure 6e in the revised manuscript).

5. Please include *Ucp1* expression level in the gonadal data in Supplemental Figure 2b.

Response: We measured mRNA levels of *Ucp1* in gonadal WAT, however these results were not included in the manuscript because the expression was below the detection limit.

6. Data in Figure 4 indicate inhibition of class I HDACs during early differentiation stage could cause adipocyte browning. It'd be interesting to know if inhibition of class I HDACs or special inactivation of HDAC3 in differentiated mature adipocytes could cause browning as well? This is potentially a clue of whether HDAC inhibition-mediated browning is more relevant to beige adipogenesis or thermogenic activation of "dormant" white adipocytes under physiological conditions.

Response: We have previously performed this experiment that was not included in the manuscript. Our results indicate that the strongest effects of MS-275 are in cells differentiated in the presence of the class I HDAC inhibitor. However, we also detected increased expression of *Ucp1* and of genes of fatty acid oxidation in terminally differentiated adipocytes treated with MS-275, though to a lesser extent. These results suggest that browning induced by class I HDAC inhibition is probably mediated also via thermogenic activation of dormant white adipocytes. We could not recapitulate this result in cells silenced with HDAC3 shRNAs, because the infection with adenoviral vehicle was not successful in terminally differentiated adipocytes.

Reviewer #4

General comment:

The manuscript by Ferrari et al. evaluates the role of histone deacetylase 3 (HDAC3) on the function of white adipose tissue (WAT) and energy metabolism. The authors found that a biological function of HDAC3 WAT is to mediate WAT browning. The deletion of Hdac3 in adipose tissue altered the metabolic signature of WAT and was concluded to occur by increasing the flux of carbon from the mitochondria to chromatin, which results in increased acetylation of enhancers in Pparg and Ucp1, key components mediating browning and thermogenesis. The authors have investigated an important problem in the epigenetic regulation of adipose function and energy metabolism, and have reached novel conclusions which might provide novel therapeutic targets for metabolic disease. However, the impact of HDAC3 on adipose physiology is modest. Moreover, certain conclusions could be better supported with better evidence.

Major comments:

1. In the in vivo model with deletion of Hdac3 in adipose tissue, Hdac3 was clearly reduced in BAT (supplementary Fig.1). With the effect on browning of WAT, one might expect an

effect on BAT-mediated thermogenesis, since the KO mice shows better endurance upon cold exposure. Although the authors have examined the mRNA levels of gene expression in BAT, the impact on function of BAT could be further explored to confirm that indeed BAT is not accounting for the increased thermogenesis under the cold conditions, such as by biochemical quantification of lipids in the tissue, adipocyte size and et al.

Response: We expanded the panel of genes analyzed in BAT, including also some white-selective markers (Supplementary figure 3e in the revised manuscript).

We also analyzed gene expression profile and histology of BAT in mice exposed to cold, and we found that the thermogenic program is not enhanced in BAT in response to cold in Hdac3 KO mice vs. floxed mice (Supplementary Figure 3h in the revised manuscript).

2. As the lack of impact on WAT in HFD-fed mice, one might question whether HFD itself affects browning in WAT through HDAC3?

Response: We demonstrated that the establishment of Hdac3 KO phenotype is related to a metabolic reprogramming that involves several pathways (glucose and lipid metabolism). This metabolic rewiring is strictly linked to events occurring at selective chromatin regions (e.g. *Ppara*, *Pparg*, *Ucp1*), highlighting that histones acetylation is an integral part of metabolism linked to transcriptional regulation. In this scenario, high fat diet feeding perturbs the equilibrium of this metabolic flux leading to a different phenotypic outcome. Some of the key metabolic events induced by Hdac3 ablation are in fact blunted by exposure to dietary lipid overload (see figure... and discussion). It should be noted that the loss of browning in inguinal fat of HDAC3 KO mice, made obese with high fed diet, seems to recapitulate the impaired thermogenesis typically observed in overweight/obese individuals as well as in rodent models of obesity. Altogether, these considerations suggest a possible role of HDAC3 in the adaptive response of adipose tissue to changes in external temperature, revealing a novel function of HDAC3 in the physiology of adipose tissue.

3. The authors claimed that there is increased lipogenesis and lipolysis in WAT, which composed a futile cycle in both animal models and in vitro cell models, mainly based on mRNA data. To support these claims, additional functional analysis are required such as the examination the lipogenesis in WAT and cells by radioactive labeling. The increased lipolysis could be better served by monitor the kinetics of the release of glycerol and fatty acids in culture media (for cells or tissue explants).

The levels of glycerol and fatty acids in plasma would likely provide some insights.

Response: To provide evidence of the flux of acetyl groups, we have performed isotope labeling experiments with ¹³C-palmitate to confirm fatty acid β-oxidation occurring in response to Hdac3 shRNA.

In cells incubated with ¹³C-palmitate (uniformly labeled), we also measured the formation of ¹³C (+2) palmitate that can be originated only from the incorporation of ¹³C (+2) acetyl-CoA, as a read out of *de novo* fatty acid synthesis.

Furthermore, we measured glycerol in culture media of C3H10T1/2 adipocytes silenced with scramble/HDAC3 shRNAs, and free fatty acids in serum of floxed and Hdac3KO mice.

Minor:

1. The size of the adipocytes could be quantified in a more meaningful way by (μm^2) instead of pixels

Response: We did the conversion from pixels to μm^2 .

2. Fig 1a - The western blot needs to be improved to the level of publication quality. Equal loading across the lanes is essential.

Response: We improved the quality of western blots included in the manuscript.

Reviewers' comments:

Reviewer #1 (Remarks to the Author):

The revision of the present manuscript by Ferrara et al. entitled 'Histone deacetylase 3 is a molecular brake of the metabolic rewiring that sustains browning of white adipose tissue' improved the manuscript vastly. However, before the manuscript can be accepted, a few issues have to be resolved.

1) I do not agree with the requirement of mice for ChIP-sequencing – if there is enough material for quantitative PCR, there should be enough material for library preparation. Successful library preps can be obtained from as little as 1ng of DNA after chromatin IP. Having said this, as the authors demonstrate changes on genes affected by Hdac3 knock-out I would not insist on sequencing though it would help to draw more detailed conclusions on the specificity of the change.

2) My major concerns lies not within the data, but with the interpretation of the connection between metabolism (levels of acetyl-CoA) and the changes observed on H3K27ac at the tested enhancers. I would suggest to tone down this part of the manuscript and focus more on the transcriptional changes introduced by Hdac3 ablation. The reasoning for this is explained below. Although the authors demonstrate that the level of acetyl-CoA is increased in the Hdac3 knock-out at a set of enhancers, no effect on global acetylation levels were observed. Equally important, no change of flux at the H3K27ac was observed. The authors cite work from Wellen et al, in which ATP-Citrate Lyase was demonstrated to be important for the generation of acetyl-CoA and its flux to histones. In contrast to Hdac3 knock-out/-down Wellen et al. observed strong effects on pan-histone acetylation. Thus, it is very difficult to distinguish if the observed changes in expression (and its correlating phenotype) stem from Hdac3 ablation induced transcriptional changes in the first place, or whether the change in metabolism induces these transcriptional changes. To shed more light on these questions, the authors should measure acetyl-CoA levels and pan-acetylation of histones in the Hdac3/ACLY double knock-down to rule out that ACLY simply acts in a dominant negative fashion. Would overexpression of ACLY alone induce the observed phenotypic changes? Finally, the quantification of nuclear ACLY is not very convincing as the protein is not detected in many floxed cells, in other words, is the detection of nuclear ACLY dependent on the expression level?

3) In line with the idea that the changes in acetylation level are due to transcriptional changes – the authors only tested genes that were upregulated upon depletion of Hdac3. As there is no net increase in this mark, the authors should test downregulated genes to determine if H3K27ac decreases.

4) It is very difficult to judge the significance of the functional annotation of the gene expression data. I suppose this is reflected in the enrichment score. I would like to see p-values for the individual functional classes (e.g. as $-\log_{10}(\text{p-value})$).

Minor points:

p. 7, line 173: What is specific mRNA? Is it cDNA generated by polydT and then amplified using gene-specific primers?

p. 8, line 187: total RNA – what was sequenced – polyA enriched RNA or ribo-zero RNA?

p. 13, line 303: GEO accession number should be given

p. 20, line 486: data refers to Supplementary Fig. 8a

p. 22, line 538: what are peculiar chromatin regions (specific?)

Reviewer #2 (Remarks to the Author):

Overall, the authors have done a lot of work for the revision and have done a good job in addressing the concerns raised. The manuscript is greatly improved from the initial submission. There are a few minor issues that still need clarification before publication.

1. It is interesting that HDAC3 deletion has a much stronger effect on WAT than BAT. It would be nice to see a comparison of expression levels of HDAC3 in the two tissues, perhaps along with the comparison of other genes that is shown in Fig 1A.

2. The presentation of the data in Supplementary Fig. 10c and d is hard to follow. For example, in part c, the peptide that encompasses residues 27-40 appears to be almost entirely acetylated, and about 10% enriched from palmitate carbon. However, in part d, it looks like Lysine 27 acetylation is about 10-20% acetylated and almost entirely enriched from palmitate carbon. It is not clear how these data align- or perhaps I misunderstand what is being presented? I suggest presenting the data in a way that makes clear the total acetylation of a given residue or peptide, and the fraction of this that is enriched.

3. I think that the same data on Acly is presented both in Figure 3d and 4e? Is there a difference between these two experiments? It seems like the same experiment but different sets of genes are presented in different figures- authors may consider consolidating gene expression data for clarity.

4. I would suggest checking whether absolute quantification of the different fatty acids in Fig 4a is warranted/ accurate. Authors are reporting that C16:1 (palmitoleate) is the most abundant fatty acid, with palmitate substantially less abundant. However, this goes against a large body of literature- in control animals at least, palmitate should be the most abundant fatty acid species and palmitoleate a small pool. This raises questions about the accuracy of the methods used. Please double check the accuracy of what is being reported and consider whether it would be more appropriate to report relative rather than absolute quant of each fatty acid between the genotypes.

5. Figure 4 L and M, Using a U-13C-palmitate tracer, M+2 would be expected as the major product rather than M+1. It would be helpful to present all isotopologues of citrate, as well as acetyl-Coa M+1 and M+2. If M+1 is the major product, how this occurs should also be made more clear in the text. I see that the potential explanation is included Supplementary Figure 9, but this was a little hard to find and deserves a little more discussion since it is not necessarily intuitive.

Reviewer #3 (Remarks to the Author):

The authors have made a serious attempt to address the reviewers' concerns with the addition of new data/experiments. Some questions have been left unanswered; however, the response is acceptable. A minor suggestion would be to incorporate some of these responses into the discussion section in an effort to highlight some of the unresolved issues that need to be addressed going forward. The discussion section as it stands seems to largely repeat the description of results.

Reviewer #4 (Remarks to the Author):

My comments have been addressed. This will be an important paper for the metabolism/epigenetics field because it extends these mechanism to the study of adiposity.

NCOMMS-16-01290A "Histone deacetylase 3 is a molecular brake of the metabolic rewiring that sustains browning of white adipose tissue" – Point-by-point Response to Reviewers

On behalf of all authors of the manuscript n. NCOMMS-16-01290A "***Histone deacetylase 3 is a molecular brake of the metabolic rewiring that sustains browning of white adipose tissue***", I would like to thank all reviewers for providing us with additional comments to further improve our manuscript.

Based on these new comments, we modified the manuscript in the effort to clarify the few points raised in the second revision of the manuscript that I am now submitting for further evaluation.

Following is the point-by-point response to the few concerns raised by each reviewer.

Reviewer #1 (Remarks to the Author):

The revision of the present manuscript by Ferrara et al. entitled 'Histone deacetylase 3 is a molecular brake of the metabolic rewiring that sustains browning of white adipose tissue' improved the manuscript vastly. However, before the manuscript can be accepted, a few issues have to be resolved.

1) I do not agree with the requirement of mice for ChIP-sequencing – if there is enough material for quantitative PCR, there should be enough material for library preparation. Successful library preps can be obtained from as little as 1ng of DNA after chromatin IP. Having said this, as the authors demonstrate changes on genes affected by Hdac3 knock-out I would not insist on sequencing though it would help to draw more detailed conclusions on the specificity of the change.

Response: As mentioned in our previous point-by-point response to reviewers, ChIP-seq from adipose tissue is really challenging. Although theoretically a few ng of IPed DNA should be enough for NGS, in real life we know by experience that IPed DNA for qPCR may not perform equally well for NGS. A number of additional variables must be properly taken into account. Having said so, we appreciate that this reviewer did not insist with ChIP-seq experiment as it would have been technically difficult with WAT.

2) My major concerns lies not within the data, but with the interpretation of the connection between metabolism (levels of acetyl-CoA) and the changes observed on H3K27ac at the tested enhancers. I would suggest to tone down this part of the manuscript and focus more on the transcriptional changes introduced by Hdac3 ablation. The reasoning for this is explained below. Although the authors demonstrate that the level of acetyl-CoA is increased in the Hdac3 knock-out at a set of enhancers, no effect on global acetylation levels were observed. Equally important, no change of flux at the H3K27ac was observed. The authors cite work from Wellen et al, in which ATP-Citrate Lyase was demonstrated to be important for the generation of acetyl-CoA and its flux to histones. In contrast to Hdac3 knock-out/-down Wellen et al. observed strong effects on pan-histone acetylation. Thus, it is very difficult to distinguish if the observed changes in expression (and its correlating phenotype) stem from Hdac3 ablation induced transcriptional changes in the first place, or whether the change in metabolism induces these transcriptional changes. To shed more light on these questions, the authors should measure acetyl-CoA levels and pan-acetylation

of histones in the Hdac3/ACLY double knock-down to rule out that ACLY simply acts in a dominant negative fashion. Would overexpression of ACLY alone induce the observed phenotypic changes? Finally, the quantification of nuclear ACLY is not very convincing as the protein is not detected in many floxed cells, in other words, is the detection of nuclear ACLY dependent on the expression level?

Response: We carefully thought about this remark and we understand the concern of this reviewer about the possibility “to distinguish if the observed changes in expression (and its correlating phenotype) stem from Hdac3 ablation induced transcriptional changes in the first place, or whether the change in metabolism induces these transcriptional changes” and the request to tone down this part of the manuscript. We apologize for not being clear enough on a critical concept, therefore we revised the section of the discussion on this issue. At this regard, our view is that in order to induce the observed metabolic changes in WAT, some transcriptional circuits should act in the first place as a consequence of Hdac3 ablation, otherwise it would not be possible to initiate the events leading to the new phenotype. Subsequently, the altered expression of proteins involved in metabolism leads to the observed phenotype. However, we do not rule out that, after the primary transcriptional events triggered by Hdac3 ablation (i.e., induction of *Ppara* and *Pparg*), the consequent metabolic alteration may drive the expression of other genes. Concerning the suggestion to “measure acetyl-CoA levels and pan-acetylation of histones in the Hdac3/ACLY double knock-down to rule out that ACLY simply acts in a dominant negative fashion”, we would like to note that in the discussion we inappropriately defined ACLY as a “mediator of the effects of Hdac3 ablation” and this definition was probably misleading. Now we rephrased the discussion highlighting that ACLY is essential for establishing the phenotype induced by Hdac3 ablation. Based on Wellen et al., ACLY is required to provide nuclear acetyl-CoA necessary for histone acetylation. Accordingly, co-silencing of ACLY and HDAC3 leads to a significant reduction of acetylation on selective chromatin regions (i.e., *Ppara*, *Pparg* and *Ucp1* enhancers) that is observed in cells where HDAC3 was silenced alone, as well as in adipose tissue of Hdac3 KO mice. Also, it should be mentioned that Hdac3 KO did not show differences in pan-acetylation of histones, thus it would be barely relevant to verify whether co-silencing of both ACLY and HDAC3 alters pan-acetylation of histones. Instead, the results of ChIP experiments in adipocytes co-silenced in both HDAC3 and ACLY, which have been already included in the second version of the manuscript, are in our opinion more meaningful in that they show reduced acetylation of specific chromatin regions that are relevant to the regulation of events required for establishing the observed phenotype. Finally, we believe that measuring the levels of acetyl-CoA in co-silenced adipocytes would not be informative as it is not possible to distinguish whether the measured acetyl-CoA belongs to the nuclear, cytosolic or mitochondrial pools. As for the question about the quantification of nuclear ACLY, we are a bit surprised that this comment came up in the revision of the second version of the manuscript, while this concern was not raised in the original submission of the manuscript. Nonetheless, to respond to this specific comment of Reviewer 1, on one hand it is true that in ingWAT of floxed mice ACLY protein seems to be less detected and this could be a consequence of different level of expression. On the other hand, the final outcome is that in Hdac3KO mice there is more ACLY protein in the nucleus that provides acetyl-CoA as a substrate for histone acetylation.

3) In line with the idea that the changes in acetylation level are due to transcriptional changes – the authors only tested genes that were upregulated upon depletion of Hdac3. As there is no net increase in this mark, the authors should test downregulated genes to determine if H3K27ac decreases.

Response: We considered this specific request and we would be willing to determine whether H3K27ac decreases in regulatory regions (i.e., enhancers/promoters) of downregulated genes in Hdac3 KO/KD. However, it is impossible to perform these measurements as we do not know what regions should be amplified by qPCR. Furthermore, IPed DNA from CHIP assays already shown in different figures was completely used up to amplify these enhancers regions by qPCR. Thus, in order to perform these control tests, we would have to start over from tissues that are very scarce or even breed more mice, implying that we would have to obtain a new ethical permit before proceeding (and this will require at least two months, to be optimistic). This would obviously delay significantly the possibility to submit the manuscript in a timely fashion.

4) It is very difficult to judge the significance of the functional annotation of the gene expression data. I suppose this is reflected in the enrichment score. I would like to see p-values for the individual functional classes (e.g. as $-\log_{10}(p\text{-value})$).

Response: We are sorry for the lack of clarity. The functional annotation tool DAVID highlights GO- and other annotation terms that are enriched in a given list of genes and then groups them according to the degree of their co-associated genes in functional clusters. The enrichment score shown in figure 2 represents the geometric mean (in $-\log$ scale) of the p-values of all GO- and other annotation terms belonging to the corresponding functional cluster. We have now clarified this in the method section as well as in the legend of figure 2. In addition, we have added a supplementary table (Supplementary Table 5) with all the enriched annotated terms and their individual p-values.

Minor points:

p. 7, line 173: What is specific mRNA? Is it cDNA generated by polydT and then amplified using gene-specific primers?

p. 8, line 187: total RNA – what was sequenced – polyA enriched RNA or ribo-zero RNA?

We used PolyA enriched RNA and have added this information in the method section.

p. 13, line 303: GEO accession number should be given

According to the instructions received by GEO, at this stage the accession number should only be given to the editor for reviewing purpose and not included in the manuscript. Should the manuscript be accepted for publication, then we will indicate the GEO accession number in the manuscript text.

p. 20, line 486: data refers to Supplementary Fig. 8a

Fixed

p. 22, line 538: what are peculiar chromatin regions (specific?)

We thank the reviewer for the suggestion. We changed the text accordingly.

Reviewer #2 (Remarks to the Author):

Overall, the authors have done a lot of work for the revision and have done a good job in addressing the concerns raised. The manuscript is greatly improved from the initial submission. There are a few minor issues that still need clarification before publication.

1. It is interesting that HDAC3 deletion has a much stronger effect on WAT than BAT. It would be nice to see a comparison of expression levels of HDAC3 in the two tissues, perhaps along with the comparison of other genes that is shown in Fig 1A.

Response: As suggested by this reviewer, we performed qPCR to compare the expression of HDAC3 in BAT vs. WAT. Data have been incorporated in Supplementary figure 1A and show that Hdac3 expression levels in BAT are 3-fold higher than in inguinal WAT of floxed mice.

2. The presentation of the data in Supplementary Fig. 10c and d is hard to follow. For example, in part c, the peptide that encompasses residues 27-40 appears to be almost entirely acetylated, and about 10% enriched from palmitate carbon. However, in part d, it looks like Lysine 27 acetylation is about 10-20% acetylated and almost entirely enriched from palmitate carbon. It is not clear how these data align- or perhaps I misunderstand what is being presented? I suggest presenting the data in a way that makes clear the total acetylation of a given residue or peptide, and the fraction of this that is enriched.

Response: We apologize for not being clear enough. The reviewer is absolutely right, we changed the text in the manuscript and the figure legend of the Supplementary Fig. 10, in order to clarify the results of the proteomic experiment. Furthermore, for clarity we also added the sequences of peptides analysed in this experiment highlighting lysine residues with red colour. The difference between part c and part d of the figure is that we are looking at the PTM of H3 in two different ways: part c refers to the methylation/acetylation status analysed on peptides; part d refers to the acetylation status analysed on the single Lysine residue 27 in the 27-40 peptide. Therefore, the acetylation and [13C] acetylation in part c represents the total modification occurring on the three Lysine residues within the 27-40 peptide, whereas the acetylation and [13C] acetylation in part d is the modification occurring only on the Lysine 27 in the same peptide.

3. I think that the same data on Acly is presented both in Figure 3d and 4e? Is there a difference between these two experiments? It seems like the same experiment but different sets of genes are presented in different figures- authors may consider consolidating gene expression data for clarity.

Response: We confirm that the same data on Acly expression is presented in Figure 3d and 4e. We reiterated this data in figure 4e to help the reader. We added a note in the legend of figure 4e to explain that the results in these two figures are from the same experiment.

4. I would suggest checking whether absolute quantification of the different fatty acids in Fig 4a is warranted/ accurate. Authors are reporting that C16:1 (palmitoleate) is the most abundant fatty acid, with palmitate substantially less abundant. However, this goes against a large body of literature- in control animals at least, palmitate should be the most abundant fatty acid species and palmitoleate a small pool. This raises questions about the accuracy of the methods used. Please double check the accuracy of what is being reported and consider whether it would be more appropriate to report relative rather than absolute quant of each fatty acid between the genotypes.

Response: We apologize with this reviewer who correctly pointed out this specific result. After reviewing all raw data (peak shapes, standard curves etc.) for the quantification of fatty acids in adipose tissue, we confirmed that results of all fatty acid quantification are technically robust: the levels of most fatty acids were reduced in Hdac3 KO mice. We only noticed something unusual in the quantification of palmitoleate. After reviewing these analyses, we decided to remove the results of palmitoleate to eliminate any uncertainty regarding this fatty acid. We believe that even after taking the data on palmitoleate the conclusions of our story do not change.

5. Figure 4 L and M, Using a U-13C-palmitate tracer, M+2 would be expected as the major product rather than M+1. It would be helpful to present all isotopologues of citrate, as well as acetyl-Coa M+1 and M+2. If M+1 is the major product, how this occurs should also be made more clear in the text. I see that the potential explanation is included Supplementary Figure 9, but this was a little hard to find and deserves a little more discussion since it is not necessarily intuitive.

Response: As correctly pointed out by Reviewer 2, acetyl-CoA +2 is more abundant than acetyl-CoA +1 (+1 in scrambled cells 0.25% MDV/ μ g of proteins vs +1 in shHDAC3 cells 0.31% MDV/ μ g of proteins, $p=0.016$; +2 in scrambled cells 0.57% MDV/ μ g of proteins vs. +2 in shHDAC3 cells 0.53, $p=0.507$). The lack of increased acetyl-CoA +2 in shHDAC3 cells could be explained by considering that this is the substrate for *de novo* fatty acid synthesis which is used as soon as it becomes available from fatty acid β -oxidation. In fact, palmitate +2 increases in shHDAC3 cells (see figure 4n).

As for citrate, we were able to detect the +1, +2, +4 and +5. Among these, citrate +1 and +4 were the most abundant (and this is why we decided to show only the two most abundant isotopologues), whereas the other two were at least 10-fold lower and their levels did not differ between scrambled and shHDAC3 cells (+1 in scrambled cells 0.285% MDV/ μ g of proteins vs +1 in shHDAC3 cells 0.152% MDV/ μ g of proteins, $p<0.001$; +2 in scrambled cells 0.023% MDV/ μ g of proteins vs +2 in shHDAC3 cells 0.029% MDV/ μ g of proteins $p=0.288$; +4 in scrambled cells 0.743% AUC vs +4 in shHDAC3 cells 1.030% MDV/ μ g of proteins $p=0.0012$; +5 in scrambled cells 0.029% MDV/ μ g of proteins vs. +5 in shHDAC3 cells 0.039 MDV/ μ g of proteins, $p=0.073$). We added these isotopologues to figure 4l and 4m respectively as suggested and we modified the text of the manuscript accordingly.

Reviewer #3 (Remarks to the Author):

The authors have made a serious attempt to address the reviewers' concerns with the addition of new data/experiments. Some questions have been left unanswered; however, the response is acceptable. A minor suggestion would be to incorporate some of these responses into the discussion section in an effort to highlight some of the unresolved issues that need to be addressed going forward. The discussion section as it stands seems to largely repeat the description of results.

Response: We are glad that this Reviewer appreciated the new version of the manuscript. To further ameliorate the manuscript, we revised the discussion and removed reiterations from the description of results and, more importantly, we tried to address some of the questions that according to this reviewer were not addressed completely.

Reviewer #4 (Remarks to the Author):

My comments have been addressed. This will be an important paper for the metabolism/epigenetics field because it extends these mechanism to the study of adiposity.

Response: We thank this reviewer for the positive response to our revised manuscript and would like to thank her/him for the useful suggestions that helped improve the manuscript.

REVIEWERS' COMMENTS:

Reviewer #2 (Remarks to the Author):

I appreciate the reviewer's clarifications on the remaining issues. I now recommend publication.